# On-the-Fly Adaptation to Quantization: Configuration-Aware LoRA for Efficient Fine-Tuning of Quantized LLMs

**Rongguang Ye[1]    Ming Tang[1]***    **Edith C. H. Ngai[2]**
[1]Southern University of Science and Technology
[2]The University of Hong Kong

## Abstract

As increasingly large pre-trained models are released, deploying them on edge devices for privacy-preserving applications requires effective compression. Recent works combine quantization with the fine-tuning of high-precision LoRA adapters, which can substantially reduce model size while mitigating the accuracy loss from quantization. However, edge devices have inherently heterogeneous capabilities, while performing configuration-wise fine-tuning for every quantization setting is computationally prohibitive. In this paper, we propose CoA-LoRA, a method that dynamically adjusts the LoRA adapter to arbitrary quantization configurations (i.e., the per-layer bit-width choices of a pre-trained model) without requiring repeated fine-tuning. This is accomplished via a configuration-aware model that maps each configuration to its low-rank adjustments. The effectiveness of this model critically depends on the training configuration set, a collection of configurations chosen to cover different total bit-width budgets. However, constructing a high-quality configuration set is non-trivial. We therefore design a Pareto-based configuration search that iteratively optimizes the training configuration set, yielding more precise low-rank adjustments. Our experiments demonstrate that, unlike the state-of-the-art methods that require fine-tuning a separate LoRA adapter for each configuration, CoA-LoRA incurs no additional time cost while achieving comparable or even superior performance to those methods.

## 1 Introduction

With the rapid growth of parameter scale, large language models (LLMs) have demonstrated increasingly strong capabilities across a wide range of applications (Zhang et al., 2022; Touvron et al., 2023; Liu et al., 2024a; Workshop et al., 2022; Ye & Tang, 2025). Nevertheless, their model size makes deployment on edge devices impractical. A common solution is to first quantize the pre-trained model and then fine-tune it using Low-Rank Adaptation (LoRA) (Hu et al., 2022), allowing model compression while maintaining performance (Dettmers et al., 2023; Guo et al., 2024; Xu et al., 2024). A key factor in this process is the quantization configuration, which specifies the bit-widths of layers in the model and thereby determines the overall compression level. However, existing methods are typically designed for a single fixed configuration and thus cannot generalize across diverse configuration settings. In practice, this limitation becomes critical because edge devices, ranging from smartphones to laptops, require support for diverse compression levels. Consequently, a single LoRA adapter is insufficient to deliver consistent performance across all possible configurations. Although fine-tuning a separate LoRA adapter for each configuration is possible, this can be extremely time-consuming.

In this work, we aim to address the challenge of adjusting LoRA adapters to arbitrary quantization configurations in an efficient manner. This is achieved through a configuration-aware model that learns to associate each configuration with its corresponding adapter adjustments. This goal raises two key challenges. First, directly mapping from each configuration to the full set of LoRA parameters is intractable, as the output space of the mapping is prohibitively large. Second, the

---

*Corresponding author.

effectiveness of this model hinges on the quality of the training configuration set. Yet, constructing such a configuration set is nontrivial: uniform bit-width assignments typically fail to account for the heterogeneous sensitivity of different layers, resulting in suboptimal results. To address these challenges, we propose CoA-LoRA, a configuration-aware model that integrates two key techniques: (i) a lightweight mapping from each layer's quantization configuration to a compact low-rank adjustment, which effectively reduces the dimensionality of the mapping output and enables parallel adjustment across layers; and (ii) a configuration set search based on a Pareto-based Gaussian process (Williams & Rasmussen, 1995), which refines the training configuration set and guides more accurate LoRA adjustment. By integrating these two techniques, CoA-LoRA produces high-quality LoRA adapters that generalize well across heterogeneous devices, eliminating the need for repeated fine-tuning when encountering new configurations.

In summary, we make the following contributions:

- We introduce CoA-LoRA, a configuration-aware method that generates lightweight low-rank adjustments to LoRA adapters based on layer information and quantization settings, enabling LoRA adapters to be adjusted to any configuration without separate fine-tuning.
- We propose a quantization configuration search technique that identifies high-performing configurations across a wide range of bit-widths and leverages them to guide the optimization of the configuration-aware model.
- Empirical experiments show that CoA-LoRA efficiently serves all quantization configurations with a single trained configuration-aware model, avoids the need for separate fine-tuning of each configuration, and achieves superior performance on most datasets with accuracy gains ranging from $1.74\%$ to $8.89\%$ over state-of-the-art methods.

## 2 RELATED WORK

### 2.1 LOW-RANK ADAPTATION OF LARGE LANGUAGE MODELS

Low-Rank Adaptation (LoRA) (Hu et al., 2022) is a widely used parameter-efficient fine-tuning method. It operates by approximating weight updates with low-rank matrices during fine-tuning, which are then merged back into the pretrained weights for inference. Owing to its low inference latency and strong adaptation capability, LoRA has become a widely adopted approach for fine-tuning LLMs (Kopiczko et al., 2023; Zhang et al., 2024; Qin et al., 2024). Several variants of LoRA have been proposed to improve its flexibility and performance. For instance, AdaLoRA (Zhang et al., 2023) incorporates an importance-aware rank allocation strategy to assign ranks according to the significance of each layer, while DoRA (Liu et al., 2024b) further decomposes the low-rank matrices into magnitude and direction components to approximate full-parameter fine-tuning. Beyond these variants, LoRA has also been explored in federated learning to reduce communication overhead between clients and servers (Singhal et al., 2025; Busacca et al., 2024), and more recently applied to restore the performance of quantized LLMs (Qin et al., 2024; Dettmers et al., 2023). Despite these successes, LoRA adapters remain sensitive in practical deployment, especially when models are quantized. This challenge highlights the need for approaches that can retain LoRA's effectiveness across diverse quantization settings.

### 2.2 WEIGHT QUANTIZATION OF LARGE LANGUAGE MODELS

Weight quantization, which quantizes the model weights while keeping activations in full precision, is a widely used and practical approach for deploying LLMs under memory and compute constraints. Standard round-to-nearest (RTN) quantization (Yao et al., 2021) is one of the most widely used techniques, where weights $\boldsymbol{w}$ are quantized as $\boldsymbol{w} \approx s \times \text{clamp}\left(\left\lfloor \frac{\boldsymbol{w}}{s} \right\rceil; -2^{b-1}, 2^{b-1} - 1\right)$ with scaling factor $s = \frac{\max(\boldsymbol{w})}{2^{b-1}-1}$ and bit-width $b$. NormalFloat (NF) (Dettmers et al., 2023) is a 4-bit floating-point format that employs non-uniform quantization via a lookup table to approximate the original floating-point values. Despite their advantages, RTN may suffer noticeable performance drop at extremely low bit-widths (Shao et al., 2024); NF initially employed uniform granularity, assigning the same bitwidth to every layer, which restricted fine-grained layer-wise adjustments and limited potential performance gains (Zhou et al., 2025). To alleviate this, several advanced low-bit quantization methods have been proposed. For example, GPTQ (Frantar et al., 2023) leverages second-order

information of the quantization error for precise 3- or 4-bit weight quantization, while AWQ (Lin et al., 2024) and OWQ (Lee et al., 2024) allocate quantization precision based on weight importance. Nevertheless, even with these techniques, quantized LLMs typically experience performance drops compared to pretrained LLMs. QLoRA (Dettmers et al., 2023) addresses this issue by applying LoRA technique to fine-tune quantized weights, partially restoring performance. Building on this idea, LQ-LoRA (Guo et al., 2024) further restores performance by initializing the LoRA adapter using iterative singular value decomposition (SVD), which provides a better starting point for fine-tuning. However, in realistic scenarios involving large-scale and heterogeneous devices, performing LoRA fine-tuning for every quantization configuration becomes computationally infeasible. This limitation motivates approaches that can adjust a single LoRA adapter across multiple quantization settings, enabling efficient deployment of quantized LLMs without repeated fine-tuning.

### 2.3 LoRA Adapter Generation

Early works such as P-diff families (Wang et al., 2024) and G.pt (Peebles et al., 2022) pioneered the use of diffusion models to generate network parameters. More recently, CONDP-DIFF (Jin et al., 2024) applies diffusion models to generate LoRA parameters in multi-task learning scenarios. While effective for coarse-grained tasks, it struggles with fine-grained tasks. To address this limitation, ICM-LoRA (Shao et al., 2025) leverages in-context learning, and LoRA-Gen (Xiao et al., 2025) improves both efficiency and performance using Mixture-of-Experts (MoE) (Lepikhin et al., 2021). However, these methods generally rely on expert LoRA and well-trained LoRA parameters, which are costly to obtain for quantized LLMs. In contrast, our work aims to adjust LoRA adapters in a lightweight and effective manner, restoring model performance under different quantization settings.

## 3 Motivation

Before introducing our method, we first highlight two key challenges in fine-tuning quantized LLMs: (i) individually training LoRA adapters across multiple bit-widths is highly time-consuming, and (ii) a single shared LoRA adapter cannot achieve consistently optimal performance across different quantization configurations.

To illustrate these challenges, Fig. 1 compares QLoRA and Shared-LoRA on the SST-2 task (Wang et al., 2019), using RoBERTa-Large (Liu et al., 2019). QLoRA fine-tunes a dedicated adapter for each quantization setting (2.5–4 bits), yielding strong accuracy but requiring fine-tuning effort that grows linearly with the number of quantized models. Shared-LoRA eliminates this cost by training a shared adapter across all settings, but the gain in efficiency comes with a large drop in accuracy.

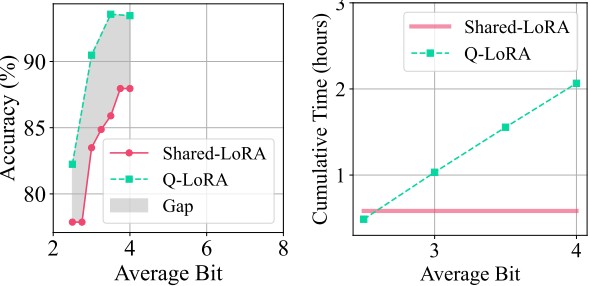

Figure 1: Accuracy gap (left) and performance comparison of cumulative fine-tuning time (right) on the SST-2 task from the GLUE benchmark using RoBERTa-Large model.

These observations naturally motivate the following question: *can we design a method that efficiently adjust the LoRA adapter across different quantization settings without repeated fine-tuning?*

## 4 CoA-LoRA: Efficient Fine-Tuning of Quantized LLMs

### 4.1 Overview of CoA-LoRA

As shown in Fig. 2, CoA-LoRA consists of two complementary techniques: (I) **Configuration-Aware LoRA Adjustment**, which trains a model $\theta$ to generate configuration-specific adjustments to LoRA matrices. A key challenge is that learning a mapping into the entire LoRA adapter space for each configuration significantly increases both the learning burden and the size of $\theta$. To address this, we generate LoRA adjustments for each layer in parallel, which substantially reduces both model size and training effort. (II) **Quantization Configuration Search and Filtering**, which identifies

**(I) Configuration-Aware LoRA Adjustment**

Configuration Embedding Space $\xrightarrow{\theta}$ Adjustment Space $\xrightarrow{\text{Evaluate}}$ Objective Space

**(II) Quantization Configuration Search and Filtering**

Figure 2: CoA-LoRA workflow: optimizing both the quantization configurations and the configuration-aware model to achieve maximum accuracy at any given bit-width.

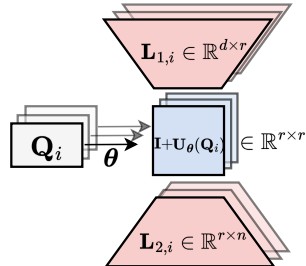

**(I) Configuration-Aware LoRA Adjustment**

Figure 3: Illustration of configuration-aware LoRA adapters with parallel adjustment. The configuration-aware model $\theta$ generates adjustment matrices $\mathbf{I} + \mathbf{U}_{\theta}(\mathbf{C}_i)$ from the quantization configuration $\mathbf{C}_i$ in parallel, where $\mathbf{I}$ denotes the identity matrix.

high-quality and diverse configurations used for training $\theta$. The main challenge lies in evaluating the quality of a configuration set and optimizing over the high-dimensional discrete configuration space. We tackle this via a Pareto-based Gaussian process combined with finite-difference gradient approximation to efficiently optimize the training configuration set.

## 4.2 CONFIGURATION-AWARE LoRA ADJUSTMENT

Given a pre-trained weight matrix $\mathbf{W} \in \mathbb{R}^{d \times n}$ and a quantization configuration $\mathbf{C}$, quantization produces a quantized matrix $\widetilde{\mathbf{W}}_{\mathbf{C}} \in \mathbb{R}^{d \times n}$. To restore the quantization error, LoRA introduces two low-rank matrices, $\mathbf{L}_1 \in \mathbb{R}^{d \times r}$ and $\mathbf{L}_2 \in \mathbb{R}^{r \times n}$, where $r \ll \min\{d, n\}$. Given a fine-tuning dataset $\mathcal{D}$ and a task-specific loss $\mathcal{L}$, the optimization problem can be expressed as

$$\underset{\mathbf{L}_1, \mathbf{L}_2}{\arg\min} \, \mathcal{L}\big(\widetilde{\mathbf{W}}_{\mathbf{C}} + \mathbf{L}_1 \mathbf{L}_2; \mathcal{D}\big). \tag{1}$$

**Quantization Configuration Representation.** We adopt the NormalFloat (NF) quantization scheme (Dettmers et al., 2023) as the quantization method to determine a quantization configuration.

We choose NormalFloat (NF) primarily because its non-uniform quantization allows better preservation of small-magnitude weights compared to uniform integer-based quantization at the same bit-width, which is critical for maintaining performance in low-bit quantization.

For each layer of a LLM, the quantization configuration of NF contains five key parameters: (1) the bit size for the initial matrix $b_0$, (2) the first-level bucket size $B_0$, (3) the bit size for quantizing block-wise absmax values $b_1$, (4) the second-level bucket size $B_1$, and (5) the final bit size for casting the absmax vector $b_2$. A *layer-level quantization configuration* is thus given by the parameter set $\boldsymbol{c}_i =$

Table 1: Layer-level configuration space of quantization parameters.

| Parameter | Configuration Space |
|---|---|
| $b_0$ | $\{2, 3, 4, 8\}$ |
| $b_1$ | $\{2, 3, 4, 8\}$ |
| $b_2$ | $\{\texttt{bf16}, \texttt{fp16}, \texttt{fp32}\}$ |
| $B_0$ | $\{16, 32, 64\}$ |
| $B_1$ | $\{16, 64, 256\}$ |

$[b_{0,i}, b_{1,i}, b_{2,i}, B_{0,i}, B_{1,i}]$ for layer $i$. Concatenating the configurations of all quantized layers yields a *model-level quantization configuration* $\{\boldsymbol{c}_1, \dots, \boldsymbol{c}_N\}$, where $N$ is the number of layers to which LoRA is applied.

Under the corresponding layer-level configuration $\boldsymbol{c}_i$, layer $i$ contains a pair of low-rank matrices $\mathbf{L}_{1,i}, \mathbf{L}_{2,i}$ that need to be adjusted. According to the layer-level configuration space of parameters

listed in Table 1, the overall search space is $(4 \cdot 4 \cdot 3 \cdot 3 \cdot 3)^N$, which grows exponentially with $N$. To make this space tractable for learning, we embed each layer-level configuration $c_i$ into a learned vector $z_i$. In addition, we further embed the layer name (e.g., fully connected layers or attention projection layers) and the block index into vectors $m$ and $b$, respectively. The final embedding for layer $i$ under the $j$-th model-level configuration is given by $\mathbf{Q}_i^{(j)} \triangleq [z_i^{(j)}, m_i^{(j)}, b_i^{(j)}]$. Based on these layer embeddings, the $j$-th model-level configuration continuous embedding is defined as their concatenation $\mathbf{Q}^{(j)} = [\mathbf{Q}_1^{(j)}, \ldots, \mathbf{Q}_N^{(j)}]$. In the following, we denote the corresponding discrete quantization configuration as $\mathbf{C}^{(j)}$.

**Configuration-Aware Model.** Mapping a configuration embedding to the full set of LoRA parameters $\{\mathbf{L}_{1,i}, \mathbf{L}_{2,i}\}_{i \in [N]}$ would be prohibitively high-dimensional. To overcome this problem, we leverage the observation that most of the adaptation signal is concentrated in $\mathbf{L}_{2,i}$ (Zhu et al., 2024; Hao et al., 2024), a finding that we also observed under quantized fine-tuning (see Appendix C.1).

Motivated by this observation, we introduce a configuration-aware model $\theta : \mathbb{R}^{|\mathbf{Q}_i|} \to \mathbb{R}^{r \times r}$, which maps a layer-level configuration vector $\mathbf{Q}_i$ to a lightweight adjustment matrix $\mathbf{U}_\theta(\mathbf{Q}_i) \in \mathbb{R}^{r \times r}$. As shown in Fig. 3, each layer's low-rank matrix $\mathbf{L}_{2,i}$ is reparameterized as $(\mathbf{I} + \mathbf{U}_\theta(\mathbf{Q}_i))\mathbf{L}_{2,i}$, where $\mathbf{I}$ is the identity matrix. Given a dataset $\mathcal{D}$, let $\widetilde{\mathbf{W}}_\mathbf{C}$ denote the quantized pre-trained model weights under configuration $\mathbf{C}$. We define the adjusted model weights using a configuration-aware adjustment function:

$$\widetilde{\mathbf{W}}_\mathbf{C}^{\text{LoRA}} = \text{InsertLoRA}\Big(\widetilde{\mathbf{W}}_\mathbf{C}, \big\{\mathbf{L}_{1,i}^{(\mathbf{C})}(\mathbf{I} + \mathbf{U}_\theta(\mathbf{Q}_i))\mathbf{L}_{2,i}^{(\mathbf{C})}\big\}_{i=1}^N\Big), \tag{2}$$

where $\text{InsertLoRA}(\cdot)$ inserts each layer's LoRA adjustment into the corresponding layer. During the generation of the LoRA adjustment, our algorithm computes the residual between pretrained weights and the quantized weights under $\mathbf{C}$ and applies an SVD to derive $\mathbf{L}_{1,i}^{(\mathbf{C})}$ and $\mathbf{L}_{2,i}^{(\mathbf{C})}$.

The configuration-aware model is then optimized by minimizing the expected task-specific loss $\mathcal{L}$ over a set of configurations $\mathcal{C}$:

$$\theta = \arg\min_\theta \mathbb{E}_{\mathbf{C} \in \mathcal{C}}\Big[\mathcal{L}\Big(\widetilde{\mathbf{W}}_\mathbf{C}^{\text{LoRA}}; \mathcal{D}\Big)\Big] \tag{3}$$

For practical implementation, we initialize the configuration set $\mathcal{C}$ with 50 configurations uniformly sampled between 2.25 and 7.25 bits (see Appendix A.2 for details).

Instead of predicting all LoRA parameters at once, which is prohibitively high-dimensional ($\sum_{i=1}^N d_i \times r + r \times n_i$), the configuration-aware model generates an $r \times r$ adjustment for each layer in parallel, significantly reducing the output dimensionality and enabling efficient learning.

### 4.3 PARETO-BASED QUANTIZATION CONFIGURATION SEARCH AND FILTERING

**Pareto-Based Surrogate Modeling.** To improve the quality of the initialized configuration set $\mathcal{Q}$ for training the configuration-aware model, we iteratively update it through a Pareto-based search. Each candidate configuration is evaluated according to two inherently conflicting objectives: task-specific performance $f_1$ and average bit-width $f_2$, as higher performance typically requires higher precision. We therefore formulate a bi-objective optimization problem for each configuration $\mathbf{C}$:

$$\min_\mathbf{C} \ \mathbf{f}(\mathbf{C}) = [f_1(\mathbf{C}), f_2(\mathbf{C})]^\top . \tag{4}$$

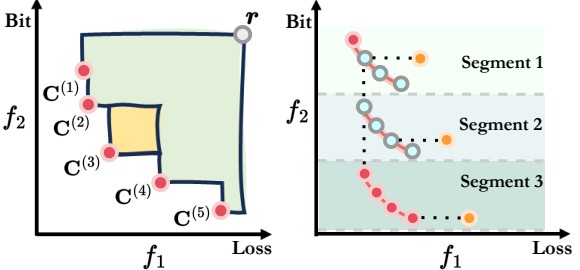

Figure 4: Illustration of the Hypervolume Improvement (left) and the Segmented Pareto Front (right). In the right figure, red points indicate the Pareto-optimal configurations, blue points are the configurations preserved in the final set after segmentation, and yellow points are discarded suboptimal configurations.

The computation of $f_2$ is provided in the Appendix (Eq. 12). By identifying the *Pareto-optimal configurations*—those for which no other configuration improves one objective without degrading the

other—we obtain a set of configurations that is both high-performing and diverse, forming the selected training configuration set $\mathcal{C}$. Projecting these configurations onto the objective space $(f_1, f_2)$ defines the *Pareto front*. To evaluate the quality of a set of trade-off configurations, we use the hypervolume (HV) metric (Zitzler & Thiele, 1999). Given a reference point $r$, the hypervolume $\mathcal{H}_r$ measures the area dominated by the Pareto front. In Fig. 4 (left), the combined green and yellow areas indicate the hypervolume formed by the five trade-off configurations with respect to $r$, where a larger HV value indicates a better Pareto front.

Direct gradient-based optimization of the configuration set $\mathcal{C}$ is infeasible because computing $f_2$ of a model-level quantization configuration in Eq. (4) involves non-differentiable quantization operations and computationally expensive forward passes. To address this challenge, we employ Bayesian optimization, which treats the model's task-specific performance $f_1(\mathbf{C})$ as a black-box function and uses a Gaussian process (GP) to guide the search. In particular, we model $f_1(\mathbf{C})$ using a GP:

$$\hat{f}_1(\mathbf{C}) \sim \mathcal{G}(m(\mathbf{C}), k(\mathbf{C}, \mathbf{C}')), \tag{5}$$

where $m(\mathbf{C})$ is the mean function, $k(\mathbf{C}, \mathbf{C}')$ is a kernel (e.g., RBF), and $\mathbf{C}, \mathbf{C}' \in \mathcal{C}$.

To select a configuration that maximally contributes to the hypervolume of the current configuration set, we use the Expected Hypervolume Improvement (EHVI):

$$\arg\max_{\mathbf{C}} \alpha_{\text{EHVI}}(\mathbf{C}) = \mathbb{E}_{\hat{f}_1(\mathbf{C}) \sim \mathcal{G}} \left[ \text{HVI}(\mathbf{f}(\mathbf{C}), \mathcal{C}) \right], \tag{6}$$

where $\text{HVI}(\mathbf{f}(\mathbf{C}), \mathcal{C}) = \mathcal{H}_{\mathbf{r}}(\mathcal{C} \cup \{\mathbf{f}(\mathbf{C})\}) - \mathcal{H}_{\mathbf{r}}(\mathcal{C})$ measures the potential hypervolume increase contributed by $\mathbf{C}$. For example, in Fig. 4 (left), the yellow area indicates the HVI of $\mathbf{C}^{(3)}$.

**Finite-Difference Guided Optimization.** Problem (6) is challenging to solve because the EHVI function does not admit an analytical gradient in the high-dimensional discrete configuration space. Therefore, we approximate the gradient using finite differences

$$\frac{\partial \alpha_{\text{EHVI}}}{\partial \mathbf{C}_i} \approx \frac{\alpha(\mathbf{C} + \delta \mathbf{e}_i) - \alpha(\mathbf{C} - \delta \mathbf{e}_i)}{2\delta}, \quad \forall i \in \{1, \dots, N\}, \tag{7}$$

where $\mathbf{C}_i$ denotes the embedding of the $i$-th layer, $\mathbf{e}_i$ is the unit vector along the $i$-th layer embedding, and $\delta$ is the step size (set to 1 in practice). Given the approximate gradients, we optimize each model-level configuration $\mathbf{C}^{(j)} \in \mathcal{C}$ through an iterative coordinate search. At iteration $t$, for the $j$-th configuration we select the layer-level coordinate $i^*$ (here $i^*$ depends on $j$) with the largest gradient magnitude: $i^* = \arg\max_i |\frac{\partial \alpha_{\text{EHVI}}}{\partial \mathbf{C}_i^{(j)}}|$. The model-level configuration is then updated along this coordinate:

$$\mathbf{C}^{(j)} \leftarrow \mathbf{C}^{(j)} - \text{sign}\left( \frac{\partial \alpha_{\text{EHVI}}}{\partial \mathbf{C}_{i^*}^{(j)}} \right) \mathbf{e}_{i^*}. \tag{8}$$

where $\text{sign}(\cdot)$ denotes the sign function. This procedure is repeated for $T$ steps for each configuration $\mathbf{C}^{(j)} \in \mathcal{C}$. Although each step modifies only one coordinate, multiple coordinates can be updated across $T$ steps. After completing $T$ steps for all configurations, the updated configurations are collected into a new set $\mathcal{C}'$. We then evaluate $\mathcal{C}'$ to obtain $\mathbf{f}(\mathcal{C}')$ and merge it with the original configuration set $\mathcal{C}$, forming $\mathcal{C} \cup \mathcal{C}'$.

**Diversity-Preserving Pareto Filtering.** The guided search described above expands the candidate set to $\mathcal{C} \cup \mathcal{C}'$. However, this merged set inevitably contains suboptimal configurations, such as those with identical bit-widths but strictly worse accuracy than others, shown as yellow points in Fig. 4 (right). Training directly on such low-quality configurations can hinder the learning of the configuration-aware model. To address this, we introduce a diversity-preserving filtering step, which filters the configurations to maintain both Pareto optimality and a wide range of bit-widths. Specifically, we divide $\mathcal{C} \cup \mathcal{C}'$ into $U$ consecutive segments, denoted by $\mathcal{C}_1, \dots, \mathcal{C}_U$. Within each segment $u$, we define the $u$-Pareto front as

$$\mathcal{C}_{\text{Pareto}}^{(u)} = \{\mathbf{C} \in \mathcal{C}_u \mid \mathbf{f}(\mathbf{C}') \not\succ \mathbf{f}(\mathbf{C}) \text{ for all } \mathbf{C}' \in \mathcal{C}_u, \mathbf{C}' \neq \mathbf{C}\}, \tag{9}$$

where $\mathbf{f}(\mathbf{C}') \not\succ \mathbf{f}(\mathbf{C})$ means that $\mathbf{C}$ is not dominated by $\mathbf{C}'$, i.e., there exists at least one objective for which $\mathbf{f}(\mathbf{C})$ is not worse than $\mathbf{f}(\mathbf{C}')$.

Table 2: Comparison of HV, accuracy gap, and training time across four tasks. Training Times are reported in minutes (m). The best HV and accuracy gap per task are highlighted in bold. The average accuracy gap is computed relative to QLoRA, and the dash ("-") indicates the baseline itself.

| Method | Solution | QNLI | | | MNLI | | | SST-2 | | | QQP | | |
|---|---|---|---|---|---|---|---|---|---|---|---|---|---|
| | | HV | Gap | Time | HV | Gap | Time | HV | Gap | Time | HV | Gap | Time |
| QLoRA | 6 | 0.58 | - | 119m | 0.54 | - | 208m | 0.63 | - | 97m | 0.54 | - | 189m |
| LQ-LoRA | 6 | 0.59 | +2.81% | 108m | 0.57 | +8.13% | 183m | 0.64 | +1.54% | 91m | 0.54 | +0.47% | 172m |
| Shared-LoRA | 1 | 0.60 | +2.90% | 21m | 0.57 | +8.11% | 35m | 0.61 | -5.06% | 19m | 0.53 | -1.18% | 32m |
| CoA-LoRA | $\infty$ | **0.62** | **+4.34%** | 57m | **0.59** | **+8.89%** | 91m | **0.67** | **+1.74%** | 52m | **0.60** | **+7.87%** | 88m |

The configuration set $\mathcal{C}$ is updated by taking the union over all segments, i.e., $\mathcal{C} \leftarrow \bigcup_{u=1}^{U} \mathcal{C}_{\text{Pareto}}^{(u)}$.

The overall training process of CoA-LoRA is organized as a cyclic procedure that alternates between two stages. In each epoch, we first train the configuration-aware model $\theta$ on the current set of training configurations $\mathcal{C}$, and then expand and refine $\mathcal{C}$ through gradient-guided search and diversity-preserving Pareto filtering.

# 5 EMPIRICAL RESULTS

## 5.1 EXPERIMENTAL SETTING

We evaluate CoA-LoRA in two settings: (I) fine-tuning on the C4 dataset (Dodge et al., 2021), and (II) fine-tuning on several GLUE tasks, namely QNLI, MNLI, SST-2, and QQP (Wang et al., 2019). For setting (I), we use Qwen-2.5-1.5B, Qwen-2.5-3B (Qwen et al., 2025), and LLaMA-2-7B (Touvron et al., 2023), whereas for setting (II), we employ RoBERTa-Large model (Liu et al., 2019). Our implementation is available at `https://github.com/rG223/CoA-LoRA`.

**Baselines.** We compare CoA-LoRA with several baselines: QLoRA (Dettmers et al., 2023), which quantizes the pretrained model before LoRA fine-tuning; LQ-LoRA (Guo et al., 2024), which quantizes the pretrained model and initializes LoRA with SVD; GPTQLoRA, which applies GPTQ quantization followed by LoRA fine-tuning; and Shared-LoRA, which fine-tunes multiple quantized models using a single LoRA adapter. Among them, LQ-LoRA serves as the strongest baseline due to its near-optimal initialization and one-to-one fine-tuning. *Our goal is for CoA-LoRA to achieve performance comparable to LQ-LoRA and QLoRA, while avoiding repeated fine-tuning by efficiently adjusting LoRA adapters to new quantization settings.*

**Evaluation.** Following prior work (Guo et al., 2024; Dettmers et al., 2023), we evaluate model performance across GLUE's benchmark tasks using accuracy. For models fine-tuned on C4, we report perplexity on the validation set. For a fair comparison, all baseline methods are quantized in a layer-wise manner; given an average bit-width across layers, the corresponding layer-wise quantization configurations are automatically determined (see Eq. (11) in the appendix). To demonstrate the effectiveness of CoA-LoRA, we present its performance over 50 configurations spanning 2.5 to 6.25 bits. For QLoRA and LQ-LoRA, due to the high computational cost of fine-tuning—each new configuration requires a separate fine-tuning run—we only report results for 2.5, 3, 3.5, 4, and 6 bits. We use *hypervolume (HV)* (Zitzler & Thiele, 1999) to evaluate the performance curves generated by each algorithm for different bit-widths. A larger HV indicates that the algorithm achieves higher performance while covering a broader range of bit-widths. Additionally, we report the *total training time* of each algorithm, as well as the *average accuracy gap* relative to QLoRA computed over four bit-widths.

**Training Settings.** We use a rank of 64 for low-rank adapters and a learning rate of $1 \times 10^{-4}$. For LLaMA-2-7B, Qwen-2.5-1.5B, and Qwen-2.5-3B, we train and evaluate on C4 with a maximum sequence length of 1024 for 5 epochs. For GLUE, we use a maximum sequence length of 128 and train for 10 epochs. The number of segmented Pareto fronts $U$ in CoA-LoRA is set to 40.

Additional experiments are provided in Appendix C, including the impact of the number of segments $U$ and zero-shot accuracy comparisons across downstream tasks, as well as other visualizations.

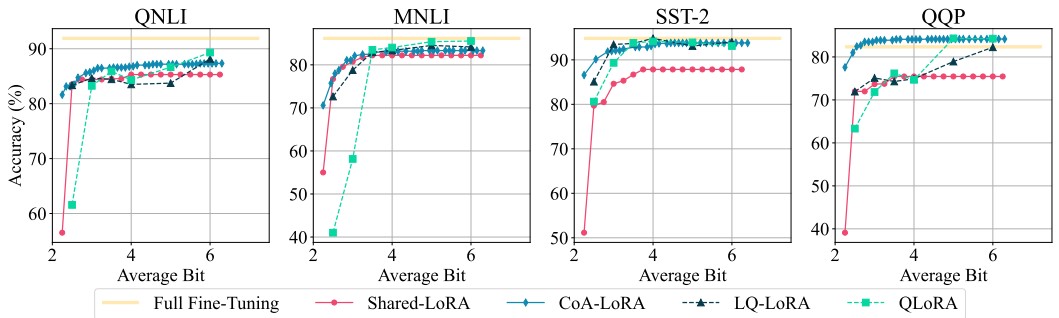

Figure 5: Comparison of accuracy across four tasks under different bit-widths.

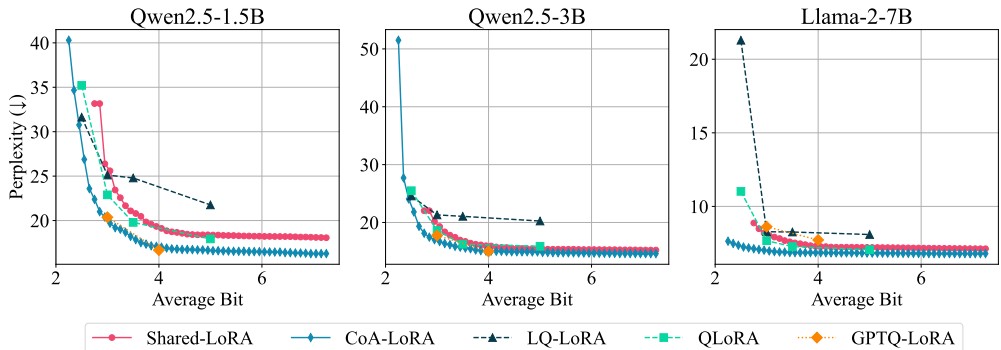

Figure 6: Performance comparison under varying bit-widths across different model sizes.

## 5.2 RESEARCH OBSERVATIONS AND EXPERIMENTS

**OBSERVATION 1:** CoA-LoRA ACHIEVES STRONG PERFORMANCE WITHOUT REPEATED FINE-TUNING.

Table 2 demonstrates that CoA-LoRA delivers strong accuracy while being more time-efficient than one-to-one fine-tuning methods (QLoRA and LQ-LoRA). This is due to its joint optimization of quantization configurations and configuration-aware model across multiple settings, which induces mutually reinforcing improvements across configurations rather than treating each configuration independently. Concretely, CoA-LoRA learns the ability to adjust LoRA adapters to *all* considered configurations in roughly one hour, whereas QLoRA and LQ-LoRA require 20–40 minutes of fine-tuning *per* configuration, so their total fine-tuning time increases proportionally with the number of configurations. Shared-LoRA reduces the total fine-tuning time by training one shared set of adapters across multiple quantized settings, but this efficiency comes with degraded accuracy (for example, accuracy gaps of $-5.06\%$ on SST-2 and $-1.18\%$ on QQP). Fig. 5 shows that CoA-LoRA maintains leading performance across various average bit-widths. Taken together, these results show that CoA-LoRA attains the favorable combination of (I) competitive or superior accuracy relative to state-of-the-art methods, and (II) substantially lower fine-tuning time compared to one-to-one fine-tuning methods, while avoiding the pronounced accuracy instability seen in Shared-LoRA.

**OBSERVATION 2:** CoA-LoRA MAINTAINS ITS EFFECTIVENESS WHEN APPLIED TO LLMS OF VARYING SIZES.

To assess scalability, Fig. 6 presents results on three pretrained models ranging from 1.5B to 7B parameters, with $N = 196, 252, 224$ layers, respectively. Across all sizes, we observe consistent trends: CoA-LoRA not only outperforms Shared-LoRA but also achieves accuracy comparable to or surpassing other state-of-the-art methods. Notably, GPTQLoRA shows curves close to CoA-LoRA on Qwen2.5-1.5B and Qwen2.5-3B, but this requires one-to-one fine-tuning at each bit-width. As model size and the number of configurations grow, this cost scales linearly, whereas CoA-LoRA achieves strong results with a single training process, demonstrating its scalability to larger LLMs.

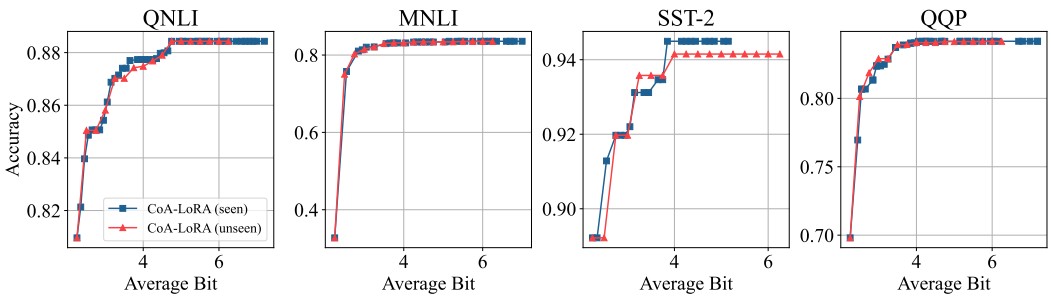

Figure 7: Comparison of CoA-LoRA performance on training and unseen configurations.

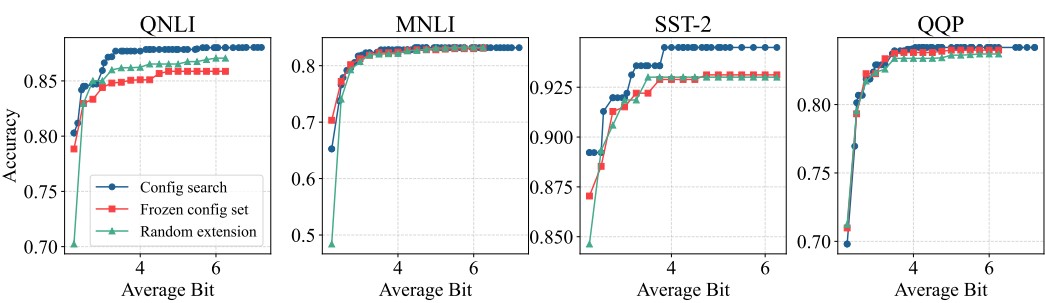

Figure 8: Effect of configuration search in CoA-LoRA.

**OBSERVATION 3:** THE LOW-RANK MATRICES ADAPTED BY CoA-LoRA EXHIBIT GENERALIZATION ACROSS UNSEEN CONFIGURATIONS.

Although Gaussian process optimization can update the quantization configuration set, CoA-LoRA is only exposed to a limited number of configurations during training. To evaluate its generalization ability to unseen configurations, in Fig. 7, we plot accuracy against bit-width for both seen and unseen configurations, which allows us to evaluate whether the model generalizes to configurations with similar average bit-widths but heterogeneous distributions. We observe that the two types of curves are highly aligned, particularly on QNLI, MNLI, and QQP, indicating that CoA-LoRA exhibits strong generalization to unseen configurations. On SST-2, while the two curves are not perfectly aligned, the discrepancy remains below $1\%$, indicating only a minor deviation. A possible explanation is that SST-2 is a simpler task compared to the others, as it does not involve reasoning across sentence pairs. The model captures less diverse patterns, which results in slightly weaker generalization when evaluated across different configuration distributions.

## 5.3 ABLATION AND SENSITIVITY ANALYSIS

**Effect of Configuration Search.** Fig. 8 illustrates the impact of applying Gaussian process–based optimization to the configuration set. To enable a more informative comparison, we further introduce a *random extension* baseline, where each epoch randomly adds ten new quantization configurations to the set. This comparison leads to two key findings. (I) Configuration search enhances the ability of the configuration-aware model to adjust low-rank matrices. Compared with the no–configuration-search setting, we observe substantial gains on QNLI and SST-2, reaching nearly $2\%$. Improvements on MNLI and QQP are smaller but remain consistent, indicating that all tasks benefit from optimization, though to varying degrees. (II) Configuration search supplies high-quality candidate configurations that effectively strengthen the configuration-aware model. In contrast, random extension achieves performance comparable to the no–configuration-search baseline but remains substantially below that of configuration search. This outcome suggests that increasing the configuration set with arbitrary configurations is ineffective—performance gains depend on introducing high-quality candidates obtained via guided optimization (i.e., Pareto-based configuration search).

Table 3: Hypervolume (HV) and accuracy gap measured at ranks 32, 64, and 128 for four tasks. The best-performing value is highlighted in bold. The dash ("-") indicates the baseline itself.

| Metric | Method | QNLI | | | MNLI | | | SST-2 | | | QQP | | |
|---|---|---|---|---|---|---|---|---|---|---|---|---|---|
| | | $r=32$ | $r=64$ | $r=128$ | $r=32$ | $r=64$ | $r=128$ | $r=32$ | $r=64$ | $r=128$ | $r=32$ | $r=64$ | $r=128$ |
| HV | QLoRA | 0.573 | 0.593 | 0.573 | 0.545 | 0.566 | 0.534 | 0.634 | 0.640 | 0.636 | 0.516 | 0.572 | 0.514 |
| | LQ-LoRA | 0.591 | 0.586 | 0.581 | 0.565 | 0.575 | 0.563 | 0.644 | 0.648 | 0.644 | 0.523 | 0.569 | 0.514 |
| | Shared-LoRA | 0.453 | 0.605 | 0.602 | 0.574 | 0.571 | 0.579 | 0.611 | 0.656 | 0.612 | 0.527 | 0.584 | 0.527 |
| | CoA-LoRA | **0.619** | **0.629** | **0.623** | **0.590** | **0.577** | **0.591** | **0.669** | **0.674** | **0.668** | **0.596** | **0.598** | **0.602** |
| Acc. Gap | QLoRA | - | - | - | - | - | - | - | - | - | - | - | - |
| | LQ-LoRA | +6.22% | -3.13% | +5.23% | +5.95% | **+2.73%** | +12.75% | **+2.84%** | **+0.91%** | +2.29% | +2.68% | +0.25% | -2.55% |
| | Shared-LoRA | -15.93% | -0.44% | +5.69% | +8.93% | -0.62% | **+13.80%** | -6.14% | -0.57% | -4.76% | +1.31% | -0.03% | +2.64% |
| | CoA-LoRA | **+6.99%** | **+1.66%** | **+6.87%** | **+9.60%** | -0.63% | +6.87% | +2.36% | +0.58% | **+2.46%** | **+10.37%** | **+0.67%** | **+11.87%** |

Table 4: Accuracy comparison under integer mixed-precision quantization (per-layer choices: `int2`, `int3`, `int4`, `int8`). Columns correspond to methods, and values within each cell are reported in the order of tasks: QNLI / MNLI / SST-2 / QQP.

| Avg. Bit | QLoRA | LQ-LoRA | Shared-LoRA | CoA-LoRA |
|---|---|---|---|---|
| 3 | 0.8537 / 0.8153 / 0.8291 / 0.7353 | 0.8552 / 0.7397 / **0.9208** / **0.7777** | 0.7772 / 0.6618 / 0.8199 / 0.7311 | **0.8616** / 0.8099 / 0.8589 / 0.7621 |
| 4 | 0.8828 / 0.8491 / 0.9116 / 0.7469 | 0.8762 / 0.7913 / 0.9185 / 0.7952 | 0.8762 / 0.7632 / 0.8486 / 0.7626 | **0.8841** / 0.8441 / **0.9254** / **0.8223** |
| 5 | 0.8859 / 0.8554 / 0.9105 / 0.7501 | 0.8761 / 0.8481 / **0.9369** / 0.8000 | 0.8777 / 0.7674 / 0.8383 / 0.7403 | **0.8863** / 0.8533 / 0.9346 / **0.8160** |
| 6 | 0.8833 / 0.8466 / 0.9174 / 0.7441 | 0.8744 / 0.8484 / **0.9323** / 0.7918 | 0.8790 / 0.7704 / 0.8440 / 0.7443 | **0.8925** / **0.8558** / 0.9323 / **0.8214** |

**Performance under Different Ranks.** Table 3 reports the hypervolume (HV) and accuracy gap across different ranks. CoA-LoRA consistently achieves the best HV values, indicating that it scales well with the output dimension of the configuration-aware model. In terms of accuracy gap, Shared-LoRA underperforms QLoRA in 7 out of 12 cases, whereas CoA-LoRA surpasses QLoRA in 11 out of 12 cases. Importantly, QLoRA and LQ-LoRA require a separate fine-tuning run for each configuration, while CoA-LoRA reaches comparable or superior results with only a single training process. This advantage comes from both optimizing quantization configurations and training across multiple settings, producing synergistic gains.

**Comparison under Integer Mixed-Precision Quantization.** Table 4 reports results for integer mixed-precision quantization. CoA-LoRA consistently achieves strong performance across the four tasks, attaining the best results in 10 out of 16 task–bit combinations. Notably, it maintains competitive accuracy even at lower bit-widths (e.g., 3 bit), whereas other methods such as QLoRA and LQLoRA tend to degrade under aggressive quantization. Across tasks, CoA-LoRA also exhibits more stable improvements as bit-width increases, while Shared-LoRA and QLoRA show more variability between datasets. These observations suggest that CoA-LoRA's robustness and versatility, demonstrating that its advantages extend beyond floating-point settings to integer-based settings.

## 6 CONCLUSION

In this work, we present CoA-LoRA, a configuration-aware approach that enables on-the-fly adjustment of low-rank adapters for arbitrary quantization configurations. Experiments show that CoA-LoRA consistently outperforms state-of-the-art methods, achieving accuracy gains of $1.74\%$–$8.89\%$ over QLoRA across four GLUE tasks, and HV improvements of $2\%$–$7\%$ compared to LQ-LoRA. Importantly, CoA-LoRA allows real-time adaptation of low-rank adapters to arbitrary configurations without additional fine-tuning, while maintaining stable performance across tasks and strong generalization to unseen configurations. These results demonstrate CoA-LoRA's efficiency and robustness for deploying large-scale LLMs under heterogeneous device capabilities.

## ACKNOWLEDGMENTS

This work was supported in part by the Guangdong Basic and Applied Basic Research Foundation under Grant 2023A1515012819 and the National Natural Science Foundation of China under Grant 62202214, and in part by the UGC General Research Fund no. 17209822 and the Innovation and Technology Commission Fund no. ITS/383/23FP from Hong Kong.

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

## THE USE OF LARGE LANGUAGE MODELS

We used large language models (LLMs) solely as a general-purpose assistant for language editing, including grammar correction and sentence polishing. LLMs did not contribute to research ideation, experimental design, analysis, or writing of original technical content. All scientific claims, experiments, and analyses in this paper are solely the work of the authors.

## A  ALGORITHM DETAILS

### A.1  CONFIGURATION-AWARE MAPPER ARCHITECTURE

The configuration-aware mapper is a conditional network that generates layer-specific adjustment matrices for LoRA parameters. The network consists of separate embeddings encoding layer and quantization information, followed by a two-layer MLP that produces a square adjustment matrix for each layer. The architecture details are summarized below:

*(i) Input Embeddings.* For each layer, several types of embeddings are constructed: **Module type:** For RoBERTa-Large: $query, key, value, output.dense, intermediate.dense$; for LLaMA and Qwen2.5: $q_{proj}, k_{proj}, v_{proj}, o_{proj}, gate_{proj}, down_{proj}, up_{proj}$. **Quantization parameters:** $b_0, b_1, b_2$ (e.g., `fp16`, `bf16`, `fp32`), $B_0, B_1$. **Block position:** the block index.

*(ii) Embedding Dimension.* Each embedding has a dimensionality of $d = 8$.

*(iii) MLP Mapping.* The concatenated embeddings form the input to a two-layer MLP with Layer-Norm and SiLU activation. The MLP has an input dimension of $7 \cdot d$, where the factor 7 corresponds to the concatenation of seven embeddings: one for the module type, five for quantization parameters, and one for the block position. The MLP has a hidden dimension of 128 and an output dimension of $r^2$, which is reshaped into an $r \times r$ adjustment matrix. The output is reshaped into an $r \times r$ adjustment matrix.

*(iv) Output Adjustment.* The adjustment matrix is scaled by a learnable parameter and added to the identity matrix to stabilize the transformation.

The base LoRA adapters are constructed by performing SVD on the difference between the full-precision and quantized weights. Further details can be found in Appendix A.2.

Adjustment matrices are generated independently for each layer, enabling on-the-fly and parallel computation for all layers. We list the trainable parameters of the configuration-aware mapper in Table A.1. The results show that the configuration-aware mapper accounts for less than 0.5% of the total parameters.

Table A.1: Parameter breakdown for different models: total model parameters / LoRA parameters (rank = 128) / configuration-aware mapper parameters.

| Model | Parameters | Percentage |
|---|---|---|
| RoBERTa-Large | 414,934,577 / 60,845,617 / 2,121,265 | 100% / 14% / 0.5% |
| Qwen2.5-1.5B | 1,693,553,729 / 149,839,425 / 2,121,281 | 100% / 8% / 0.1% |
| Qwen2.5-3B | 3,327,528,513 / 241,589,825 / 2,121,281 | 100% / 7% / 0.06% |
| Llama-2-7B | 7,060,352,577 / 321,936,961 / 2,121,281 | 100% / 4% / 0.03% |

### A.2  GENERATION OF THE INITIAL QUANTIZATION CONFIGURATION SET

Let $\{\mathbf{W}^{(i)}\}_{i \in [N]}$ denote the set of $N$ LoRA weight matrices to be adjusted. The specific layers that can be adjusted are listed in Table A.2. Inspired by (Guo et al., 2024), we introduce a binary matrix $\mathbf{X} \in \{0, 1\}^{N \times \kappa}$, where $\kappa$ denotes the number of quantization configuration variables. In our work, we consider five variables (see Table 1).

Our goal is to initialize a quantization configuration $\mathbf{C}_{|\mathbf{X}}$ with an average bit-width of $b$. To this end, we first approximate each weight matrix $\mathbf{W}$ via a low-rank decomposition using Singular Value

Decomposition (SVD):

$$\mathbf{W} - \widetilde{\mathbf{W}}_{\mathbf{X}} = \mathbf{U}\mathbf{\Sigma}\mathbf{V}^\top. \tag{10}$$

Let $\mathbf{L}_{1|\mathbf{X}} = \mathbf{U}_r\sqrt{\mathbf{\Sigma}_r}$ and $\mathbf{L}_{2|\mathbf{X}} = \sqrt{\mathbf{\Sigma}_r}\mathbf{V}_r^\top$, where $\mathbf{U}_r$, $\mathbf{\Sigma}_r$, and $\mathbf{V}_r$ correspond to the first $r$ singular components of $\mathbf{W} - \widetilde{\mathbf{W}}_{\mathbf{X}}$. We then obtain the binary $\mathbf{X}$ by solving the following 0-1 constrained optimization problem:

$$\begin{aligned}
\min_{\mathbf{X}} \quad & \left\| \mathbf{W} - \left(\widetilde{\mathbf{W}}_{\mathbf{X}} + \mathbf{L}_{1|\mathbf{X}} + \mathbf{L}_{2|\mathbf{X}}\right) \right\|_F, \\
\text{s.t.} \quad & f_2(\widetilde{\mathbf{W}}_{\mathbf{X}}) \le \text{budget}, \\
& \sum_{j=1}^{\kappa} X_{ij} = 1, \quad \forall i \in [N],
\end{aligned} \tag{11}$$

where $f_2(\widetilde{\mathbf{W}}_{\mathbf{X}})$ denotes average bit of $\widetilde{\mathbf{W}}_{\mathbf{X}}$. It can be computed as

$$f_2(\mathbf{C}) = \frac{\sum_{i=1}^{L}(b_0 + \frac{b_1}{B_0} + \frac{b_2}{B_0 B_1}) \cdot w_i}{\sum_{i=1}^{L} w_i}, \tag{12}$$

where $w_i$ represents the number of parameters in layer $i$ and $\mathbf{C} = [b_0, b_1, b_2, B_0, B_1]$. Here, $b_2$ denotes the bitwidth of the precision type. If the type is `fp16` or `bf16`, we set $b_2 = 16$; if it is `fp32`, we set $b_2 = 32$. Intuitively, this procedure seeks a binary quantization assignment $\mathbf{X}$ that balances quantization (via the storage constraint) and reconstruction error (via the low-rank residual), providing a well-initialized configuration for subsequent LoRA fine-tuning.

We generate 50 initial quantization configurations with average bit-widths ranging from 2.25 to 7.25 in steps of 0.1 by solving Problem (11). It is worth noting that this integer programming problem can be efficiently solved using standard solvers such as Gurobi[1], taking only 3–5 seconds per instance.

Table A.2: Adjustable layers for different backbone models.

| Model | Adjustable Layers |
|---|---|
| RoBERTa-Large | [`query`, `key`, `value`, `output.dense`, `intermediate.dense`] |
| Qwen2.5-1.5B | [`q_proj`, `k_proj`, `v_proj`, `o_proj`, `gate_proj`, `down_proj`, `up_proj`] |
| Qwen2.5-3B | [`q_proj`, `k_proj`, `v_proj`, `o_proj`, `gate_proj`, `down_proj`, `up_proj`] |
| Llama-2-7B | [`q_proj`, `k_proj`, `v_proj`, `o_proj`, `gate_proj`, `down_proj`, `up_proj`] |

## A.3 QUANTIZATION CONFIGURATION SELECTION FOR A GIVEN BIT-WIDTH

After CoA-LoRA has been trained, it produces a final configuration set $\mathcal{C}$, which has been iteratively refined using a Gaussian process. This set is representative of configurations across a wide range of bit-widths. Consequently, for any target bit-width $b$, we can select a quantization configuration by finding the one that is *closest* to some configuration $\mathbf{C} \in \mathcal{C}$, while ensuring that the overall average bit-width equals $b$.

Formally, this selection problem can be written as

$$\begin{aligned}
\min_{\mathbf{C}} \quad & \text{dist}(\mathbf{C}, \mathcal{C}) \\
\text{s.t.} \quad & f_2(\mathbf{C}) = b,
\end{aligned} \tag{13}$$

where $\text{dist}(\mathbf{C}, \mathcal{C}) = \min_{\mathbf{C}' \in \mathcal{C}} \|\mathbf{C} - \mathbf{C}'\|_F$ measures the distance to the nearest configuration in $\mathcal{C}$.

Intuitively, problem (13) selects the representativeness of the final configuration set: it ensures that the selected configuration is both feasible for the target bit-width and similar to the training configurations, thereby ensuring that the configuration performs well under the specified bit-width. As shown in Fig. A.1, this procedure ensures that the configuration-aware model produces LoRA adjustments that generalize effectively to new configurations with nearby bit-widths.

---

[1] https://www.gurobi.com

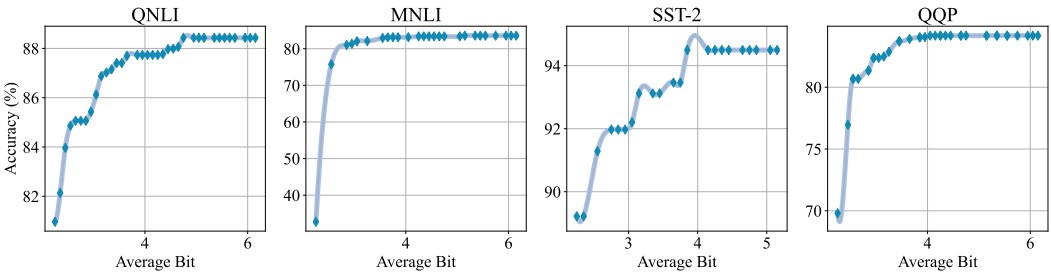

Figure A.1: Comparison of the final configuration set with arbitrary configurations across four tasks.

---

**Algorithm 1:** CoA-LoRA

**Input:** Initial quantization configurations $\mathcal{C}$, fine-tuning dataset $\mathcal{D}$, number of epochs $T_1$, number of finite-difference iterations $T_2$, and calibration data $\mathcal{D}^{ca}$

1 **for** $\mathbf{C} \in \mathcal{C}$ **do**
2     Evaluate $f_1(\mathbf{C} \mid \mathcal{D}^{ca})$ on the calibration data;
3 Let $\mathcal{E} = \{(\mathbf{C}, f_1(\mathbf{C} \mid \mathcal{D}^{ca})) \mid \mathbf{C} \in \mathcal{C}\}$;
4 Fit a Gaussian process $\mathcal{G}$ based on $\mathcal{E}$;
5 Initialize $\mathcal{C}' \leftarrow \emptyset$;
6 **for** $t_1 = 1$ **to** $T_1$ **do**
7     Sample $\mathbf{C} \sim \mathcal{C}$;
8     # Update the configuration-aware model $\theta$ ;
9     $\arg\min_{\theta} \mathbb{E}_{\mathbf{C} \in \mathcal{C}} \left[ \mathcal{L}\left( \widetilde{\mathbf{W}}_{\mathbf{C}}^{\text{LoRA}}; \mathcal{D} \right) \right]$;
10     **for** $\mathbf{C} \in \mathcal{C}$ **do**
11        **for** $t_2 = 1$ **to** $T_2$ **do**
12           # Optimize each configuration;
13           $\mathbf{C} - \text{sign}\left( \frac{\partial \alpha_{\text{EHVI}}}{\partial \mathbf{C}_{i^*}} \right) \mathbf{e}_{i^*}$;
14           # Add the optimized configuration to $\mathcal{C}'$;
15     # Obtain segmented Pareto fronts $\mathcal{C}_{\text{Pareto}}^u$ from $\mathcal{C} \cup \mathcal{C}'$, for $u = 1, \ldots, U$;
16     $\mathcal{C}_{\text{Pareto}}^{(u)} = \{\mathbf{C} \in \mathcal{C}_u \mid \mathbf{f}(\mathbf{C}) \not\succ \mathbf{f}(\mathbf{C}') \text{ for all } \mathbf{C}' \in \mathcal{C}_u, \mathbf{C}' \neq \mathbf{C}\}$;
17     # Update training configuration set;
18     $\mathcal{C} \leftarrow \bigcup_{u=1}^{U} \mathcal{C}_{\text{Pareto}}^{(u)}$;

---

## A.4 TRAINING PROCEDURE

We present the procedure for optimizing the configuration-aware model in Algorithm 1. Initially, we construct the set of quantization configurations $\mathcal{C}$ and evaluate their loss on the calibration data. In our experiments we use a single batch from the training dataset. This gives $\{f_1(\mathbf{C})\}_{\mathbf{C} \in \mathcal{C}}$. A Gaussian process is then fitted based on these evaluations. During each training epoch, we jointly optimize the configuration-aware model and update the training quantization configuration set.

## B EXPERIMENTAL DETAILS

For the computation of Hypervolume (HV) (Zitzler & Thiele, 1999), we first collect the performance metrics for each algorithm. The second objective $f_2$ can be either accuracy or perplexity depending on the experiment.

To ensure that smaller values indicate better performance, we normalize $f_2$ as follows:

$$f_2^{\text{norm}} = \begin{cases} 1 - \frac{f_2}{100}, & \text{if } f_2 \text{ is accuracy,} \\ \frac{f_2}{f_2^{\text{max}}}, & \text{if } f_2 \text{ is perplexity,} \end{cases} \tag{14}$$

Table C.1: Reference points (used for Hypervolume calculation) and block numbers for four models.

| Model | Reference Point $r$ | Block Number |
|---|---|---|
| RoBERTa-Large | (1, 1) | 24 |
| Qwen2.5-1.5B | (1, 1) | 28 |
| Qwen2.5-3B | (1, 1) | 36 |
| Llama-2-7B | (1, 1) | 32 |

Table C.2: Comparison of hypervolume (HV) and average decrease in perplexity (lower is better) relative to QLoRA across three LLMs.

| Method | Qwen2.5-1.5B | | Qwen2.5-3B | | Llama-2-7B | |
|---|---|---|---|---|---|---|
| | HV | Gap | HV | Gap | HV | Gap |
| QLoRA | 0.432 | - | 0.473 | - | 0.593 | - |
| LQ-LoRA | 0.390 | -1.87% | 0.425 | -2.74% | 0.571 | -3.22% |
| GPTQ-LoRA | 0.427 | +1.89% | 0.427 | +0.87% | 0.448 | -0.80% |
| Shared-LoRA | 0.422 | -0.88% | 0.464 | +0.42% | 0.568 | +0.30% |
| CoA-LoRA | **0.479** | **+2.97%** | **0.506** | **+1.43%** | **0.629** | **+1.25%** |

Table C.3: Zero-shot accuracy comparison across different methods on eight tasks with an average 3-bit budget. The best result for each task and average is highlighted in bold.

| Model | Method | ANLI | BoolQ | Winogrande | RTE | PiQA | ARC-Easy | ARC-Challenge | Average |
|---|---|---|---|---|---|---|---|---|---|
| Qwen2.5-3B | QLoRA | 27.34 | 66.41 | 55.47 | 59.38 | 70.34 | 50.78 | 28.12 | 51.12 |
| | LQ-LoRA | 34.38 | 67.97 | 57.81 | 57.81 | 68.75 | 52.34 | 24.22 | 51.90 |
| | Shared-LoRA | 31.25 | 64.84 | 53.91 | 54.69 | 71.88 | 52.34 | 29.69 | 51.23 |
| | CoA-LoRA | 38.28 | 62.50 | 59.38 | 55.47 | 70.31 | 55.47 | 32.03 | **53.35** |
| Llama-2-7B | QLoRA | 39.06 | 75.00 | 71.88 | 60.16 | 73.44 | 65.62 | 42.19 | 61.05 |
| | LQ-LoRA | 31.25 | 76.56 | 70.31 | 65.62 | 75.00 | 65.52 | 41.41 | 60.81 |
| | Shared-LoRA | 38.28 | 78.12 | 68.75 | 62.50 | 74.22 | 67.19 | 42.97 | 61.72 |
| | CoA-LoRA | 38.28 | 78.91 | 67.97 | 62.50 | 76.56 | 67.97 | 41.41 | **61.94** |

where $f_2^{\max}$ denotes the maximum value of $f_2$ across all algorithms. After normalization, HV is computed based on the two objectives, with values ranging from 0 to 1, where smaller values indicate better performance in both dimensions. The reference points and block numbers for each model are listed in Table C.1.

## C  ADDITIONAL EXPERIMENTS

### C.1  SIMILARITY OF LOW-RANK MATRICES ACROSS DIFFERENT BIT-WIDTH MODELS

We evaluated the correlations between $\mathbf{L}_1$ and $\mathbf{L}_2$ under different ranks and across multiple models. Figures C.1 and C.2 show the block-wise averaged similarity between $\mathbf{L}_1$ and $\mathbf{L}_2$ after quantizing RoBERTa-Large to 2, 3, 4, 5, and 6 bits using Eq. 11, followed by low-rank fine-tuning. We observe that the shared components across different configurations are mostly captured in $\mathbf{L}_2$, while the configuration-specific knowledge is primarily encoded in $\mathbf{L}_1$. This observation motivates the design of the configuration-aware model $\boldsymbol{\theta}$, where the model outputs an $r \times r$ matrix $\mathbf{U}_{\boldsymbol{\theta}}$ to directly transform $\mathbf{L}_2$ into $\mathbf{U}_{\boldsymbol{\theta}} \mathbf{L}_2$.

### C.2  COMPARISON OF ZERO-SHOT PERFORMANCE ON DOWNSTREAM TASKS

Table C.3 shows that CoA-LoRA achieves strong accuracy across diverse downstream tasks, including ANLL (Nie et al., 2020), BoolQ (Clark et al., 2019), Winogrande (Sakaguchi et al., 2021), RTE (Wang et al., 2019), PiQA (Bisk et al., 2020), ARC-Easy, and ARC-Challenge (Clark et al., 2018), with an average gain of 1.45% over the best-performing LQ-LoRA on Qwen2.5-3B. Al-

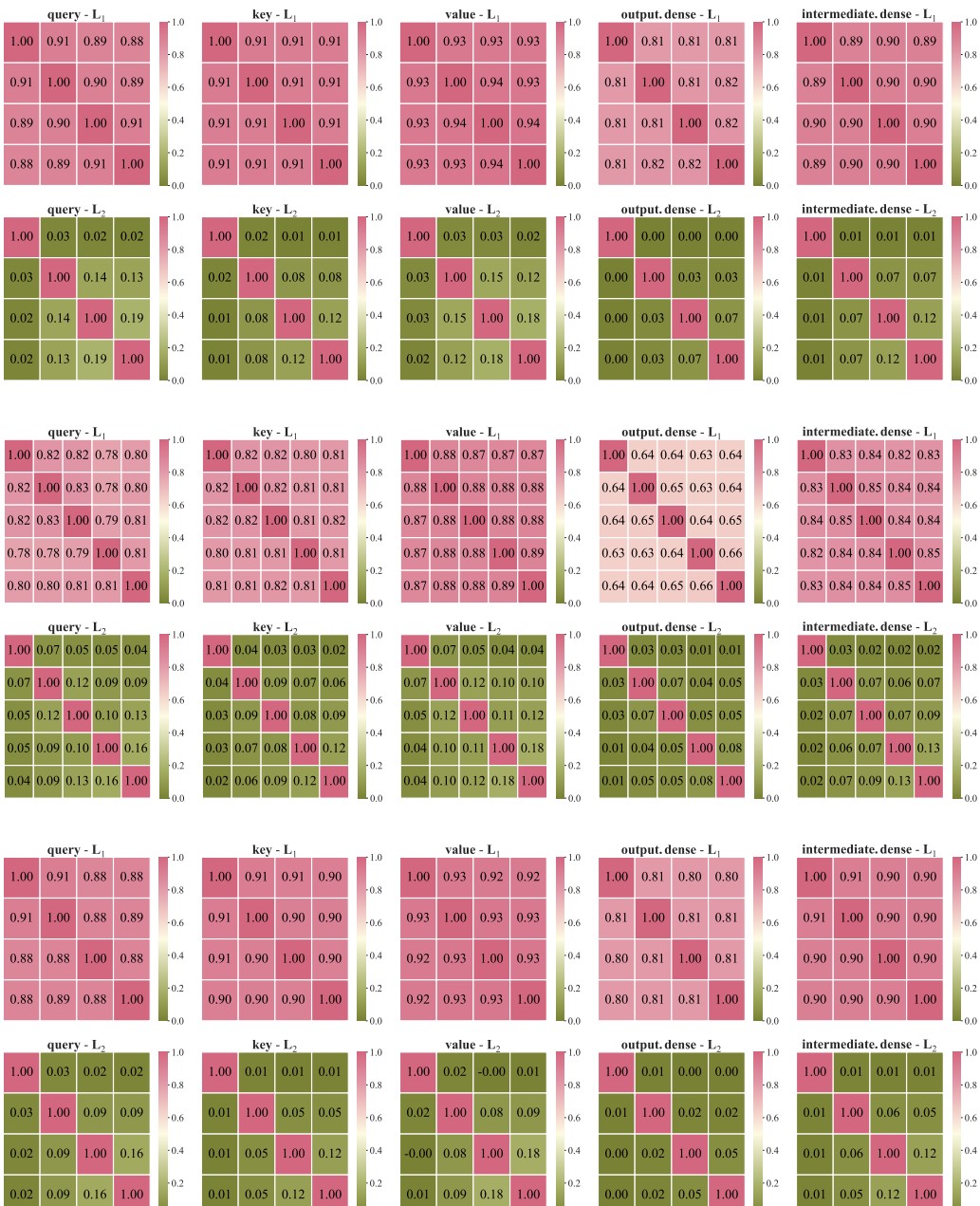

Figure C.1: Correlations of two types of low-rank matrices after LoRA fine-tuning on RoBERTa-Large for models quantized to 2, 3, 4, 5, and 6 bits, measured on the QNLI dataset. The three panels correspond to rank 32 (top), rank 64 (middle), and rank 128 (middle). Each panel shows two rows representing the block-wise averaged similarity for $\mathbf{L}_1$ and $\mathbf{L}_2$.

though fine-tuned on the C4 dataset, CoA-LoRA also maintains robust zero-shot accuracy, showing greater stability than Shared-LoRA.

## C.3 IMPACT OF THE NUMBER OF SEGMENTS $U$

Fig. C.3 compares the results under different values of $U$, where $U = 0$ corresponds to the case without segment Pareto selection. We observe that applying segment Pareto selection (i.e., $U = 20$

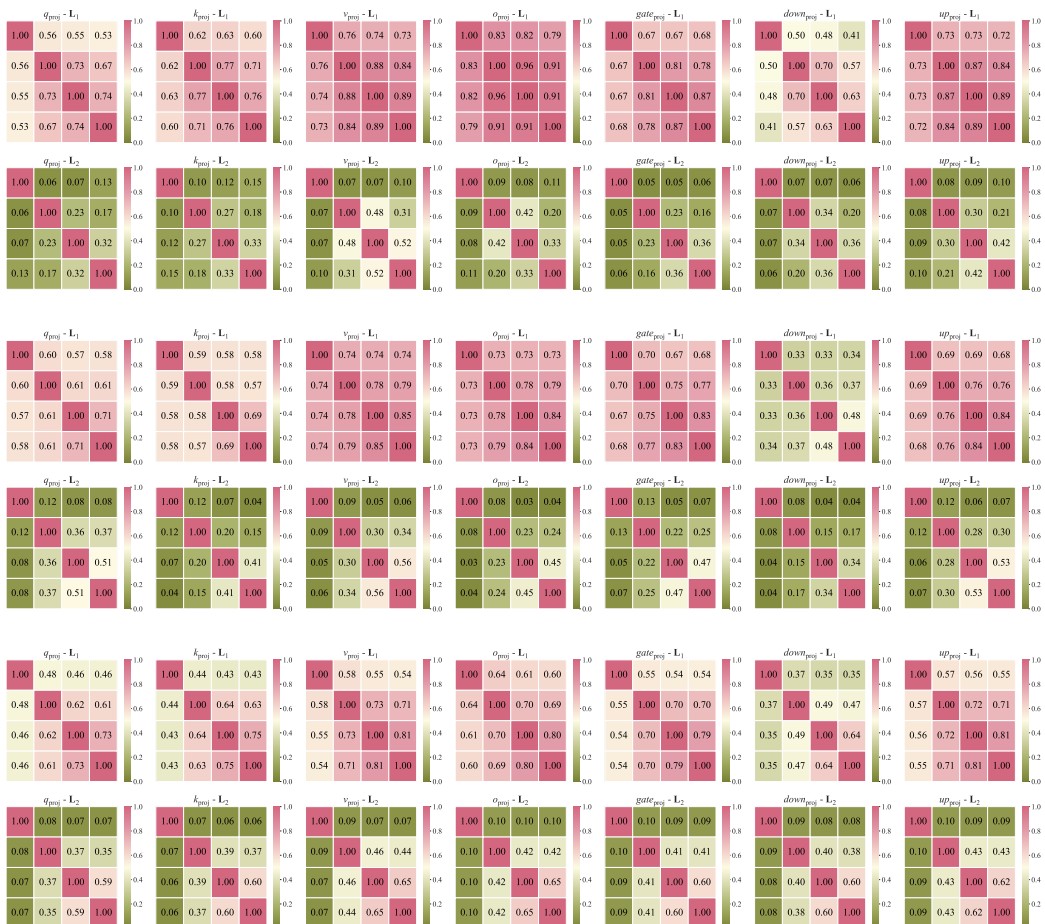

Figure C.2: Correlations of two types of low-rank matrices after LoRA fine-tuning for models quantized to 2, 3, 4, 5, and 6 bits, measured on the C4 dataset. The three panels correspond to Qwen-1.5B (top), Qwen-3B (middle), and Llama-2-7B (bottom). Each panel shows two rows representing the block-wise averaged similarity for $\mathbf{L}_1$ and $\mathbf{L}_2$.

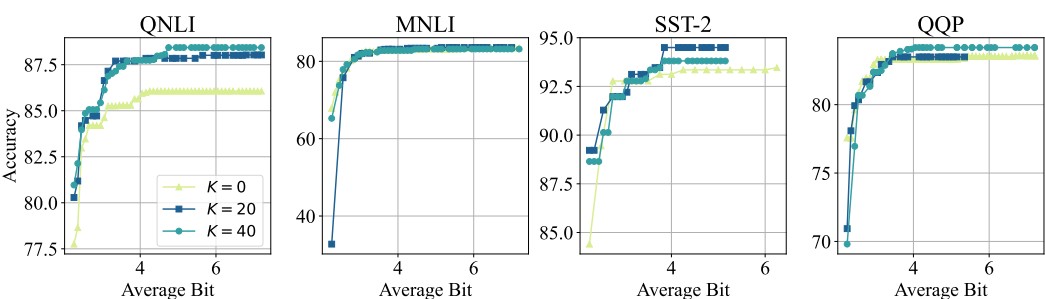

Figure C.3: Comparison of performance with different segment numbers $K$ across four tasks.

or $U = 40$) generally outperforms the baseline of $U = 0$, particularly at higher bit-widths. This indicates that segment Pareto selection helps maintain strong performance across a broad range of quantization levels. Moreover, we find that on QNLI and QQP, $U = 40$ achieves better results than $U = 20$, whereas on SST-2, $U = 40$ performs worse. A plausible explanation is that SST-2, being a relatively simple task that does not involve sentence-pair reasoning, may not benefit from too many segments, while more challenging tasks tend to require finer segmentation to capture diverse decision boundaries.

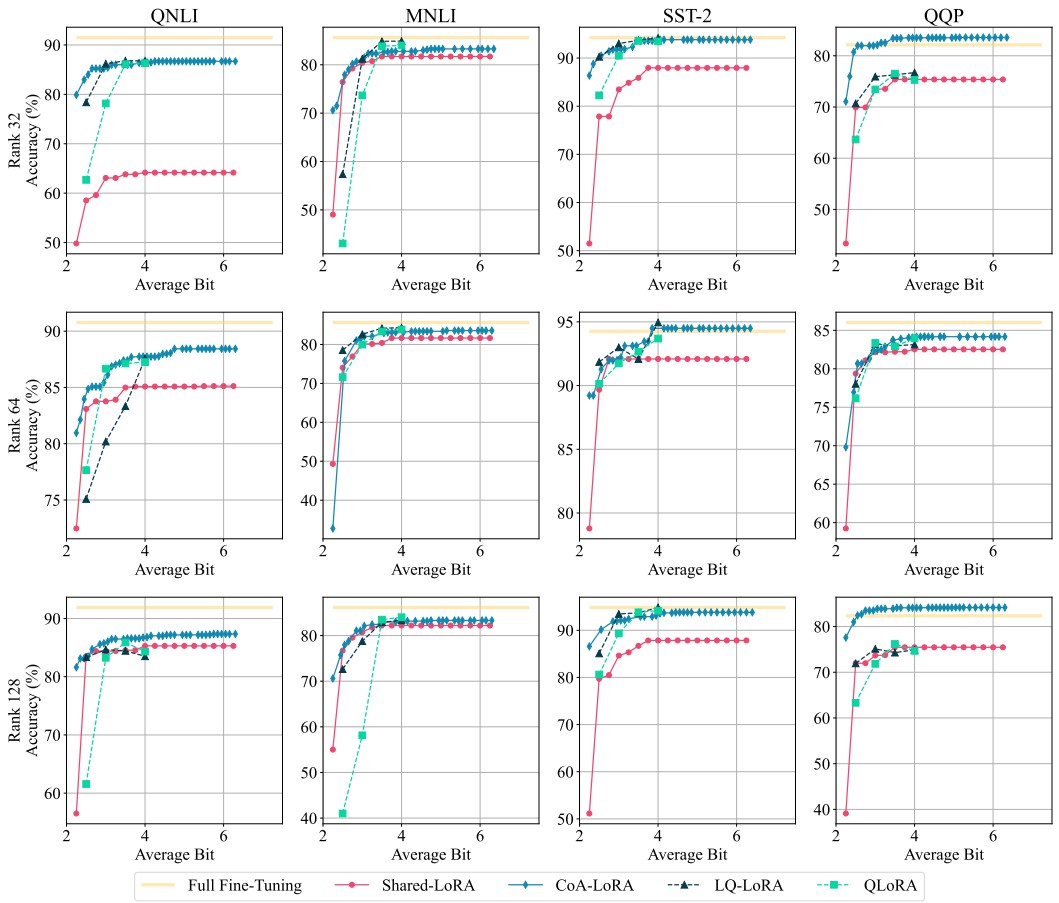

Figure C.4: Comparison of results with ranks 32, 64, and 128 across four tasks.

Table C.4: Perplexity comparison under different quantization bits (2.5, 3, 3.5, 4) on LLaMA-2-13B with rank 64. "-" denotes that GPTQ does not officially support 2.5-bit and 3.5-bit quantization.

| Avg. Bit | QLoRA | LQ-LoRA | GPTQ-LoRA | Shared-LoRA | CoA-LoRA |
|----------|-------|---------|-----------|-------------|----------|
| 2.5 | 9.11 | 9.23 | - | 8.87 | **8.65** |
| 3 | **7.48** | 8.00 | 7.95 | 8.18 | 7.58 |
| 3.5 | **7.09** | 7.52 | - | 7.59 | 7.15 |
| 4 | **6.89** | 7.32 | 7.15 | 7.28 | 6.99 |

## C.4 VISUALIZATION COMPARISON OF METHODS UNDER VARYING RANKS ON ROBERTA-LARGE

Fig. C.4 visualizes the Pareto fronts of CoA-LoRA and four baselines, showing that CoA-LoRA achieves accuracy comparable to, and often surpassing, QLoRA and LQ-LoRA, which rely on one-to-one fine-tuning after quantization.

## C.5 COMPARISON UNDER LLAMA-2-13B

We evaluated CoA-LoRA on LLaMA-2-13B across multiple quantization bits. Table C.4 reports the perplexity for different methods. CoA-LoRA achieves performance comparable to one-to-one full fine-tuning across most bit settings, including low-bit configurations (2.5–3.5 bits). These results indicate that CoA-LoRA can scale to larger models while maintaining stable and efficient adaptation to different quantization levels.

Table C.5: Performance of LoRA quantization methods on LLaMA-2-7B across different ranks (32, 64, 128) and bit-widths (2.5–4). "-" denotes that GPTQ does not officially support 2.5-bit and 3.5-bit quantization.

| Rank | Avg. Bit | QLoRA | LQ-LoRA | GPTQ-LoRA | Shared-LoRA | CoA-LoRA |
|------|----------|-------|---------|-----------|-------------|----------|
| 32 | 2.5 | 9.48 | 9.94 | - | 33.26 | **8.85** |
| | 3 | **8.28** | 8.63 | 8.63 | 8.43 | 8.76 |
| | 3.5 | 7.92 | 7.59 | - | 8.01 | **7.56** |
| | 4 | 7.70 | **7.37** | 7.73 | 7.91 | 7.49 |
| 64 | 2.5 | 11.02 | 21.29 | - | 8.87 | **8.48** |
| | 3 | **7.67** | 8.28 | 8.62 | 8.01 | 7.91 |
| | 3.5 | **7.27** | 8.26 | - | 7.52 | 7.52 |
| | 4 | **7.06** | 8.09 | 7.72 | 7.27 | 7.23 |
| 128 | 2.5 | 9.46 | 9.79 | - | 24.93 | **9.21** |
| | 3 | **7.64** | 7.86 | 8.66 | 8.78 | 7.77 |
| | 3.5 | 7.91 | **7.51** | - | 7.67 | **7.51** |
| | 4 | 7.71 | **7.33** | 7.72 | 7.54 | 7.39 |

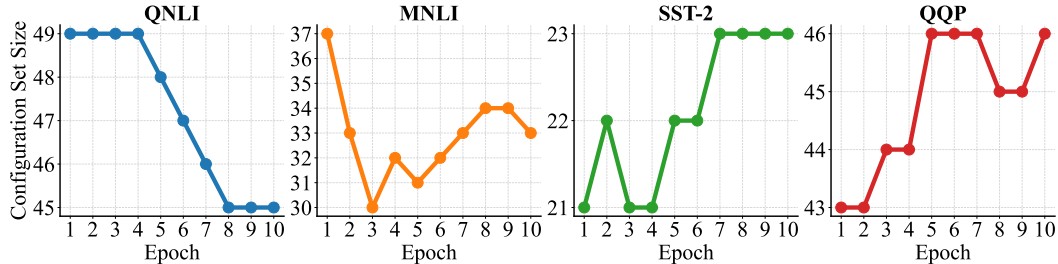

Figure C.5: Configuration set sizes of CoA-LoRA across four tasks.

## C.6 COMPARISON OF LLAMA-2-7B UNDER DIFFERENT RANKS

Table C.5 reports the performance of different methods across various ranks and bit-widths. CoA-LoRA consistently exhibits stable results across all bit settings. At rank 32, its performance remains steady, while Shared-LoRA shows extreme fluctuations at bit 2.5. Similar patterns are observed at ranks 64 and 128, where CoA-LoRA avoids the large variations exhibited by LQ-LoRA and Shared-LoRA under low bit conditions. These results indicate that CoA-LoRA preserves performance under aggressive quantization and delivers reliable behavior across different Ranks.

