# OpenReview forum: "On-the-Fly Adaptation to Quantization: Configuration-Aware LoRA for Efficient Fine-Tuning of Quantized LLMs"
_ICLR.cc/2026/Conference — ICLR 2026 Poster_

### Official Review · Reviewer_Daov · 2025-10-28

**Soundness:** 3
**Presentation:** 2
**Contribution:** 3
**Rating:** 4
**Confidence:** 4

**Summary:**

This paper proposes CoA-LoRA, a fast (on-the-fly) adaptation of LoRA for various backbone model quantization configurations. Compared to the baselines, which train separate LoRA adapters for each quantization configuration, the proposed method trains a single configuration-aware model $\theta$ that generates adjustment matrices for the LoRA adapters to adapt to a given configuration. To train the configuration-aware model, the paper proposes a Pareto-based quantization configuration search, which makes the training set (configurations) Pareto-optimal in the configuration space. The experimental results show that the proposed method achieves comparable or better accuracy with a single trained model $\theta$.

**Strengths:**

* The motivation is strong and practical. The paper proposes a method that does not require fine-tuning when changing the quantization configuration. This would save significant GPU-hours.
* The experimental results show comparable or even better accuracy compared to the baselines.
* The idea is novel and insightful. The paper proposes a meta-learning approach with a Pareto-based search algorithm for improved quantization accuracy.

**Weaknesses:**

* Most experiments were conducted on the RoBERTa-Large model (0.3B), which is hard to consider an LLM. In the six experimental results presented in the evaluation section, only a single experiment was conducted on the Qwen and LLaMA models, reporting their perplexity. Therefore, as the title suggests that the method targets LLMs, there should be more focus on actual LLMs.
* The paper is not fully reproducible. There is a lack of crucial methodological details, and no code is provided. Below is a list of information that should be included.

  * The architecture of the configuration-aware model $\theta$ is not described; the only information given is that it receives a quantization configuration and outputs an r×r matrix.
  * Is the configuration-aware model trained layer-wise or model-wise?
  * The NormalFloat datatype was originally introduced as a 4-bit datatype (NF4). How does the paper handle the k-bit NF type?
  * How are layer-wise bitwidth configurations chosen? The proposed method uses layer-wise configurations, but the paper does not explain how they are selected for evaluation.
  * A complete algorithm for using $\theta$ during quantization and training is missing. How is the backbone model trained and quantized? If the model is trained and then quantized, it consumes more memory during fine-tuning, which is precisely the issue the baseline methods aim to solve. If the model is quantized and then trained, how is the initial quantization configuration chosen, and which configuration is used for training the backbone models and LoRA adapters on the target task?
* This reviewer believes that some content in the paper should be revised.

  * line 98: The citation for NormalFloat is incorrect (Dettmers et al., 2022 → 2023).
  * line 99: NormalFloat does not use reduced exponent and fractional bits but employs non-uniform quantization based on a lookup table.
  * line 101: The cited papers do not contain information about NF suffering from accuracy degradation at very low bit-widths.
  * line 187: “NF is particularly suitable in our setting because it supports block-wise quantization at a finer granularity”: this reviewer believes NF is not the only datatype supporting block-wise quantization, as any datatype can do so. Therefore, there should be another justification for choosing NF.
  * Across the paper: QLoRA is a more accurate name than “Q-LoRA” used in the paper.

**Questions:**

* How is the average bitwidth measured? Does the model use the same configurations across layers?
* How does the model performance in Figure 5 exceed full fine-tuning?
* How is the accuracy gap measured? Since the experimental results are obtained from multiple quantization configurations, is the accuracy gap averaged?
* In the ablation study (Fig. 8), how is the performance measured without the Gaussian process? Is it replaced with random sampling and freezing? If so, this reviewer believes random sampling should be repeated during training to ensure the training set size matches that used with the Gaussian process.

---

> ### Author Response · Authors · 2025-11-18
> **Response (Part I)**
>
> We sincerely thank the reviewer for the constructive comments. Our responses are provided below:
>
> > Weakness 1: Limited evaluation
>
> Based on the reviewers’ suggestions, we have added experiments on LLaMA-2-7B under different ranks and further conducted evaluations on the larger LLaMA-3-14B. The results are presented in Tables R1 and R2. These results demonstrate that CoA-LoRA maintains consistent performance on larger models, indicating its scalability. These experiments have been incorporated into the revised manuscript. In addition, the revised manuscript includes similarity experiments of low-rank matrices on larger models (Qwen2.5-1.5B, Qwen2.5-3B, and LLaMA-2-7B) in Fig. C.2, as well as zero-shot performance evaluations on seven downstream tasks for Qwen2.5-3B and LLaMA-2-7B in Table C.3 of the manuscript.
>
> **Table R1.** Performance of LoRA quantization methods on LLaMA-2-7B across different ranks and bit-widths.
>
> | Rank | Avg. Bit | QLoRA | LQ-LoRA | GPTQ-LoRA | Shared-LoRA | CoA-LoRA |
> | --- | --- | --- | --- | --- | --- | --- |
> | 32  | 2.5 | 9.48 | 9.94 | -   | 33.26 | **8.85** |
> | 32  | 3   | **8.28** | 8.63 | 8.63 | 8.43 | 8.76 |
> | 32  | 3.5 | 7.92 | 7.59 | -   | 8.01 | **7.56** |
> | 32  | 4   | 7.70 | **7.37** | 7.73 | 7.91 | 7.49 |
> | 64  | 2.5 | 11.02 | 21.29 | -   | 8.87 | **8.48** |
> | 64  | 3   | **7.67** | 8.28 | 8.62 | 8.01 | 7.91 |
> | 64  | 3.5 | **7.27** | 8.26 | -   | 7.52 | 7.52 |
> | 64  | 4   | **7.06** | 8.09 | 7.72 | 7.27 | 7.23 |
> | 128 | 2.5 | 9.46 | 9.79 | -   | 24.93 | **9.21** |
> | 128 | 3   | **7.64** | 7.86 | 8.66 | 8.78 | 7.77 |
> | 128 | 3.5 | 7.91 | **7.51** | -   | 7.67 | **7.51** |
> | 128 | 4   | 7.71 | **7.33** | 7.72 | 7.54 | 7.39 |
>
> **Table R2.** Perplexity comparison under different quantization bits (2.5, 3, 3.5, 4) on LLaMA-3-14B.
>
> | Avg. Bit | QLoRA | LQLoRA | GPTQ-LoRA | Shared-LoRA | CoA-LoRA |
> | --- | --- | --- | --- | --- | --- |
> | 2.5 | 9.11 | 9.23 | -   | 8.87 | **8.65** |
> | 3   | **7.48** | 8.00 | 7.95 | 8.18 | 7.58 |
> | 3.5 | **7.09** | 7.52 | -   | 7.59 | 7.15 |
> | 4   | **6.89** | 7.32 | 7.15 | 7.28 | 6.99 |
>
> **Note:** It is reasonable that QLoRA and LQ-LoRA achieve higher accuracy, as they perform **one-to-one fine-tuning** tailored to specific quantization configurations. In contrast, our proposed CoA-LoRA conducts **one-for-all fine-tuning**. Therefore, achieving performance comparable to QLoRA and LQ-LoRA is expected, while our proposed method also provides the practical benefit of covering all configurations with a single fine-tuning process.
>
> > Weakness 2: Reproducibility
>
> **An anonymized version of our code has been prepared and is available at:
> https://anonymous.4open.science/r/CoA-LoRA**.
>
> We address the reviewers' comments concerning the algorithmic details as follows:
>
>
> **(i) The architecture of the configuration-aware model.** The structure of the configuration-aware model is as follows:
>
> 1. **Input Embeddings:** For each layer, several types of embeddings are constructed:
>
>   - **Module type:** For RoBERTa-Large, the module types include `query`, `key`, `value`, `output.dense`, and `intermediate.dense`; for LLaMA, they include `q_proj`, `k_proj`, `v_proj`, `o_proj`, `gate_proj`, `down_proj`, and `up_proj`.
>   - **Quantization parameters:** $b_0, b_1, b_2$ (e.g., $\texttt{fp16, bf16, fp32}), B_0, B_1$.
>   - **Block position**: the block index.
> 2. **Concatenation**: All embeddings are concatenated into a single vector representing the layer and its quantization configuration.
>
> 3. **MLP**: A two-layer MLP with a normalization layer and SiLU activation maps the concatenated vector to an output of size $r×r$ (where $r$ is the LoRA rank). The MLP produces an adjustment matrix for the LoRA parameters of the corresponding layer.
>
> 4. **Output adjustment**: The output matrix is scaled by a learnable parameter and an identity matrix is added to stabilize the transformation, resulting in the final layer-specific adjustment matrix.
>
>
> **(ii)** **Configuration-aware model training.** A configuration-aware model allocates low-rank adjustments in a layer-wise manner, while its training is performed in a model-wise level, because the forward pass and loss computation require all layers’ low-rank adjustments to be assigned before processing the training data of the downstream task.
>
> **(iii)** **K-bit NF**. Our method supports k-bit NormalFloat types by adopting the official NF implementation used in LQ-LoRA. In this implementation, each tensor is quantized in a block-wise manner: for each block, the maximum absolute value is computed and the values are mapped to a k-bit NormalFloat codebook corresponding to the chosen bit-width.
>
> Please refer to Part II for the remaining responses.

---

> ### Author Response · Authors · 2025-11-18
> **Response (Part II)**
>
> **(iv)** **Layer-wise bitwidth selection strategy**. The layer-wise quantization configurations are initialized as described in Appendix A.2. Specifically, the initialization is performed via SVD on the difference between the original and quantized weights:
>
> $$
> \mathbf{W} - \widetilde{\mathbf{W}}_{\mathbf{X}} = \mathbf{U} \mathbf{\Sigma} \mathbf{V}^\top,
> $$
>
> where $\mathbf{X} \in \{0,1\}^{N \times \kappa}$ is a binary optimization variables representing the quantization configuration, and $\kappa$ denotes the number of quantization variables. The objective is to minimize the discrepancy between the quantized and original pre-trained weights under a specified average bit-width. This optimization problem can be efficiently solved by Gurobi software to obtain a quantization configuration for any given average bit budget. The detailed mathematical formulation is provided in Eq. (11) in the appendix.
>
> Regarding which configuration is selected, our method first initializes a configuration set that spans a wide range of average bit levels, as specified by Eq. (11) in the Appendix. We then perform sampling-based optimization within this configuration set to learn an effective configuration-aware model. During this process, the configuration set is refined via finite-difference evaluation and Pareto filtering, as detailed in Section 4.3 of the paper.
>
> **(v) Algorithmic details.** The training process consists of three steps:
> *(1) Initialization.* We generate an initial configuration set that spans a wide range of average bit levels; please refer to (iv) above for the construction procedure.
>
> *(2) Configuration-aware prediction.* The configuration-aware model takes the embedding of each layer’s quantization configuration as input and learns to generate an $r×r$ low-rank adjustment for each layer. These adjustments are incorporated into the corresponding LoRA modules, which are then merged with the quantized base model.
>
> *(3) Evaluation and update.* The merged model is evaluated on a small portion of the task-specific training dataset to compute its loss, based on which the configuration-aware model is updated.
>
> **Algorithm 1 of the Appendix presents this process.**
>
> Throughout the entire process, the backbone model does not require full-precision training; All methods first quantize the pretrained backbone and then fine-tune LoRA on the resulting quantized model. Other baselines fine-tune only the LoRA modules, whereas our method trains the configuration-aware model.
>
> The selection of initial quantization configurations follows the procedure described in the response (IV): we first construct an initial configuration set, and during training, the configuration-aware model is optimized by sampling configurations from this set for the downstream task.
>
> **(vi) Justification for choosing NF.** We choose NormalFloat (NF) primarily because its non-uniform quantization allows better preservation of small-magnitude weights compared to uniform integer-based quantization at the same bit-width, which is critical for maintaining accuracy in low-bit quantization.
>
> To further validate our approach, we have also conducted experiments using mixed-integer precision quantization, where each layer's bit-width is selected from {$\texttt{int2}, \texttt{int3}, \texttt{int4}, \texttt{int8}$}, demonstrating that our configuration-aware method remains effective across both NF and mixed-integer precision schemes.
>
> **Table R3:** Accuracy comparison under integer mixed-precision quantization (per-layer choices: $\texttt{int2}, \texttt{int3}, \texttt{int4}, \texttt{int8}$). Columns correspond to methods, and values within each cell are reported in the order of tasks: QNLI / MNLI / SST-2 / QQP.
>
> | Avg. Bit | QLoRA | LQ-LoRA | Shared-LoRA | CoA-LoRA |
> | --- | --- | --- | --- | --- |
> | 3   | 0.8537 / **0.8153** / 0.8291 / 0.7353 | 0.8552 / 0.7397 / **0.9208** / 0.7777 | 0.7772 / 0.6618 / 0.8199 / 0.7311 | **0.8616** / 0.8099 / 0.8589 / 0.7621 |
> | 4   | 0.8828 / **0.8491** / 0.9116 / 0.7469 | 0.8762 / 0.7913 / 0.9185 / 0.7952 | 0.8762 / 0.7632 / 0.8486 / 0.7626 | **0.8841** / 0.8441 / **0.9254** / **0.8223** |
> | 5   | 0.8859 / **0.8554** / 0.9105 / 0.7501 | 0.8761 / 0.8481 / 0.9369 / 0.8000 | 0.8777 / 0.7674 / 0.8383 / 0.7403 | **0.8863**/ 0.8533 / 0.9346 / **0.8160** |
> | 6   | 0.8833 / 0.8466 / 0.9174 / 0.7441 | 0.8744 / 0.8484 / 0.9323 / 0.7918 | 0.8790 / 0.7704 / 0.8440 / 0.7443 | **0.8925** / **0.8558** / **0.9323** / **0.8214** |
>
> **(vii) inaccurate and unclear expression**. In the revised manuscript, we have corrected several inaccurate statements, including the naming of QLoRA (replacing “Q-LoRA”), the citation for NormalFloat (updated from Dettmers et al., 2022 to 2023), and the description of NormalFloat’s performance degradation under very low-bit settings.
>
> Please refer to Part III for the remaining responses.

---

> ### Author Response · Authors · 2025-11-18
> **Response (Part III)**
>
> > Question 1: Layer-wise average bitwidth and configuration
>
> The average bit-width is measured as
>
> $$
> f_2(\mathbf{C}) = \frac{\sum_{i=1}^{L} (b_0+\frac{b_1}{B_0}+\frac{b_2}{B_0B_1}) \cdot w_i}{\sum_{i=1}^{L} w_i},
> $$
>
> where $w_i$ is the number of parameters in layer $i$. and $\mathbf{C}=[b_0,b_1,b_2,B_0,B_1]$. Here, $b_2$ denotes the bitwidth of the precision type. If the type is $\texttt{fp16}$ or $\texttt{bf16}$, we set $b_2 = 16$; if it is $\texttt{fp32}$, we set $b_2 = 32$.
>
> For a fair comparison, our proposed CoA-LoRA and other baselines, such as QLoRA and LQ-LoRA, employ layer-wise bit quantization rather than using a uniform bit-width across all layers. The procedure for generating layer-wise configurations is described in Weakness 2 (iv), with detailed steps provided in Appendix A.2.
>
> > Question 2: Unexpected QQP results: CoA-LoRA vs. full fine-tuning
>
> Although full fine-tuning is theoretically superior to LoRA, in practice LoRA can occasionally match or even surpass full fine-tuning, depending on the task. For example, in [1], LoRA outperforms full fine-tuning on COCO Captioning (Table 2 in the original paper), and similarly on Visual Instruction Tuning tasks (see Table 3). Fig. 1 of [2] also shows that, with appropriate rank settings, LoRA can sometimes exceed full fine-tuning. Moreover, in our experiments, the baseline LQ-LoRA occasionally achieves results slightly better than full fine-tuning (Fig. 2 in the original paper).
>
> This phenomenon can be partly explained by task characteristics and model generalization. QQP is a text similarity task where some examples have subtle semantic differences, making them more challenging for the model to generalize. Full fine-tuning updates all model parameters, which can fit training data very closely, while LoRA updates only low-rank adapters. This constrained update acts as a form of implicit regularization, reducing sensitivity to subtle variations and potentially improving generalization on challenging examples. As a result, LoRA can occasionally outperform full fine-tuning on QQP.
>
> Furthermore, there are two main reasons why CoA-LoRA is more likely to outperform full fine-tuning on QQP
>
> - As shown by the similarity matrices in Fig. C1 and Fig. C2 of the appendix, $\mathbf{L}_1$ captures shared knowledge across different average quantization configurations (i.e., high similarity), while $\mathbf{L}_2$ focuses more on task-specific knowledge for each bit setting. This observation indicates that LoRA fine-tuning under different average bits exhibits similar patterns rather than being entirely unrelated tasks. CoA-LoRA, by training on a diverse set of quantization configurations, effectively leverages this shared structure, thereby enhancing the learning of shared knowledge.
>
> - High-bit configurations preserve more information from the original full-precision model, providing more reliable gradient directions for learning. Low-bit configurations can then take advantage of the parameter patterns (or structures) learned under high-bit conditions to efficiently converge to suitable LoRA parameters, rather than learning from scratch as in QLoRA and LQ-LoRA. This effect is illustrated in Fig. 5 of the revised manuscript: under low-bit settings, CoA-LoRA often outperforms QLoRA and LQ-LoRA, while in high-bit settings, QLoRA and LQ-LoRA can achieve performance comparable to CoA-LoRA. This indicates that low-bit training provides little benefit for high-bit configurations.
>
>
> [1] S. Liu, C. Wang, H. Yin, et al. Dora: Weight-Decomposed Low-Rank Adaptation. International Conference on Machine Learning. 2024.
>
> [2] D. Biderman, J. Portes, J. J. G. Ortiz, et al. LoRA Learns Less and Forgets Less. Transactions on Machine Learning Research. 2024.
>
> > Question 3: Average accuracy gap
>
> For each method, its accuracy gap is computed as the mean difference in accuracy relative to QLoRA across multiple quantization configurations.
>
>
> Please refer to Part IV for the response to the final question.

---

> ### Author Response · Authors · 2025-11-18
> **Response (Part IV)**
>
> > Question 4: The effect of training configuration set size
>
> Table R4 shows how the configuration set size in CoA-LoRA evolves over epochs under the Gaussian process, starting from an initial size of 50. These results are now included in Fig. C.5 of the Appendix.
>
> We observe that the configuration set size remains relatively stable over 10 epochs, indicating that the Gaussian process does not improve performance simply by increasing the number of configurations. Instead, it works via a survival-of-the-fittest process, selecting strong configurations and eliminating weaker ones. In Fig. 8 of the manuscript, the “No config search” setting refers to freezing the initial configuration set at size 50. Based on the reviewer's suggestion, we have changed "No config search" to "Frozen config set" to improve clarity.
>
> To further investigate whether randomly adding configurations to the set could improve performance, we added a "Random extension" experiment in the revised Fig. 8. In "Random extension" setting, we start with the same 50 initial configurations as in the "Config search" setup, and 10 configurations are randomly added at each epoch, resulting in a configuration set of size 150 at epoch 10. The results show that the performance of “Random extension” is comparable to that of “Frozen config set,” but both are inferior to our proposed “Config search” mechanism.
>
> **Table R4.** Configuration set sizes of CoA-LoRA across four tasks.
>
> | Epoch | QNLI | MNLI | SST-2 | QQP |
> | --- | --- | --- | --- | --- |
> | 1   | 49  | 37  | 21  | 43  |
> | 2   | 49  | 33  | 22  | 43  |
> | 3   | 49  | 30  | 21  | 44  |
> | 4   | 49  | 32  | 21  | 44  |
> | 5   | 48  | 31  | 22  | 46  |
> | 6   | 47  | 32  | 22  | 46  |
> | 7   | 46  | 33  | 23  | 46  |
> | 8   | 45  | 34  | 23  | 45  |
> | 9   | 45  | 34  | 23  | 45  |
> | 10  | 45  | 33  | 23  | 46  |

---

> > ### Comment · Reviewer_Daov · 2025-11-20
> >
> > This reviewer appreciates the rebuttal and acknowledges that it has addressed all previously raised concerns, including the major points related to LLM performance and reproducibility. As a minor clarification, regarding Table R3, please include NF4 since the original request was to compare INT and NF4.
> > After considering the rebuttal and the other reviewers’ feedback, I will adjust my rating accordingly.

---

> ### Author Response · Authors · 2025-11-21
>
> We are glad that we were able to address the reviewer’s concerns. We have added the NF4 experiments to Table R3, as shown below.
>
> | Avg. Bit | QLoRA                                 | LQ-LoRA                           | Shared-LoRA                       | NF4                                   | CoA-LoRA                                      |
> |:--------:| ------------------------------------- | --------------------------------- | --------------------------------- | ------------------------------------- | --------------------------------------------- |
> | 4        | 0.8828 / **0.8491** / 0.9116 / 0.7469 | 0.8762 / 0.7913 / 0.9185 / 0.7952 | 0.8762 / 0.7632 / 0.8486 / 0.7626	 | 0.8817 / 0.8448 / **0.9254** / 0.7563 | **0.8841** / 0.8441 / **0.9254** / **0.8223** |
>
>
> From the comparison of results, NF4 demonstrates competitive performance relative to QLoRA, but remains slightly inferior to CoA-LoRA.
>
> We appreciate the reviewer’s suggestion and will incorporate these experiments into the final version of the manuscript.

---

### Official Review · Reviewer_XFF9 · 2025-10-28

**Soundness:** 4
**Presentation:** 4
**Contribution:** 2
**Rating:** 6
**Confidence:** 3

**Summary:**

The authors propose a way to directly predict LoRA-weights suitable for a given quantization configuration of the base-model.
The method solves a bi-level optimization problem. A mapping from the space of possible base-model quantization mappings to LoRA-weights is learned. In order to learn this mapping, a suitable training-set is tuned at the same time as the mapping.
The resulting optimized training set along with the mapping makes it possible to find a quantization configuration setting for a given average bit-width constraint (Appendix A.2) and then immediately infer the LoRA weights that aims to recover the performance drop for that configuration setting. Consequently, it is possible to obtain LoRA-weights for a specific bit-width constraint without any necessary fine-tuning.

**Strengths:**

The paper presents an original idea, each step of the solution is well-motivated. The quality of writing is high, the paper is information-dense yet good to follow.

**Weaknesses:**

The main weakness I see with the paper is in its problem motivation. When does anyone require the ability to zero-shot LoRA weights for a new quantization configuration? From a research perspective, this problem fascinates me. From a practitioner's perspective, I don't know when I would ever need this. In deploying a model on edge devices, I can reasonably enumerate the set of accelerators I aim to target for a given model (or model-class) and therefore target the corresponding bit-width. On the other hand, a practitioner might care more about latency for a given bit-width of a specific layer - or have specific configuration constraints as far as hardware-level support for quantization configurations is concerned. I understand you choose NF, but I assume your method should generalize to other - practically relevant - formats. I see this as an opportunity to discuss and motivate your method in a wider context: We might not care only about storage, but also latency, under additional hardware constraints.

**Questions:**

Figure 7: Does "unseen" mean the configuration has never been part of $\mathcal{C}$ or does it mean it's not part of the final configuration set?

What exactly is the configuration-aware model - and does it matter? You mention "light-weight", but how exactly do you compose [z, m, b] and U?

I find a conceptual break between the fact that your search-space lies in the continuous C, whereas valid configurations are discrete and enumerable. Do you make sure that your update (8) remains within valid bounds? How precisely do you evaluate f_1(C) for a given C (EQ 4) - or equivalently: what exactly is $\tilde{W}_C$ if C lies somewhere between the embeddings of the discrete quantization configurations?

---

> ### Author Response · Authors · 2025-11-18
> **Response to Weaknesses (Part I)**
>
> We sincerely thank the reviewer for the thoughtful comments. Our responses are provided below:
>
> > Weakness: Practical motivation under hardware constraints and generality across quantization formats.
>
> The scenario raised by the reviewer—selecting a quantization configuration based on known device constraints to achieve low latency—is fully compatible with our problem setting. We discuss this in two practical deployment cases:
>
> **(1) Arbitrary unknown-device setting.**
> Our method can produce LoRA adjustments for a wide range of quantization configurations after only one fine-tuning round. With one-for-all fine-tuning, the model is prepared for deployment on a range of future user devices with various hardware and accelerator capabilities, while still allowing users to customize quantization settings based on the bit widths and memory budgets supported by their eventual hardware.
>
> **(2) Single known-device setting.**
> Given the device specifications, users can decide a preferred layer-wise quantization configuration. Our method can then generate the corresponding LoRA weights to recover accuracy under that configuration. This process provides two advantages:
>
> - **Minimal latency overhead.** Prior work [1] demonstrates that the inference latency introduced by LoRA is negligible, remaining nearly identical to the baseline without adapters. This observation aligns with the low computational overhead of LoRA in our setting.
>
> - **Flexible compression under memory pressure.** In typical deployment environments, available memory gradually decreases as more applications run. Users may thus reduce bit-width to further shrink the model. In this case, retraining LoRA for each new configuration would be costly, whereas CoA-LoRA can directly generate accuracy-restoring adapters without any retraining.
>
>
> Regarding the reviewer’s comment on generalization beyond NF, we conducted additional experiments under integer mixed-precision quantization, where each layer can choose from $\texttt{int2}, \texttt{int3}, \texttt{int4}$, and $\texttt{int8}$. The results are reported in the Table R1. CoA-LoRA consistently achieves strong performance across all four tasks, obtaining the best score in 10 out of 16 task–bit combinations. These results and a detailed analysis have been added to Section 5.3 of the revised manuscript.
>
> **Table R1:** Accuracy comparison under integer mixed-precision quantization (per-layer choices: $\texttt{int2}, \texttt{int3}, \texttt{int4}, \texttt{int8})$. Columns correspond to methods, and values within each cell are reported in the order of tasks: QNLI / MNLI / SST-2 / QQP.
>
> | Avg. Bit | QLoRA | LQ-LoRA | Shared-LoRA | CoA-LoRA |
> | --- | --- | --- | --- | --- |
> | 3   | 0.8537 / **0.8153** / 0.8291 / 0.7353 | 0.8552 / 0.7397 / **0.9208** / 0.7777 | 0.7772 / 0.6618 / 0.8199 / 0.7311 | **0.8616** / 0.8099 / 0.8589 / 0.7621 |
> | 4   | 0.8828 / **0.8491** / 0.9116 / 0.7469 | 0.8762 / 0.7913 / 0.9185 / 0.7952 | 0.8762 / 0.7632 / 0.8486 / 0.7626 | **0.8841** / 0.8441 / **0.9254** / **0.8223** |
> | 5   | 0.8859 / **0.8554** / 0.9105 / 0.7501 | 0.8761 / 0.8481 / 0.9369 / 0.8000 | 0.8777 / 0.7674 / 0.8383 / 0.7403 | **0.8863**/ 0.8533 / 0.9346 / **0.8160** |
> | 6   | 0.8833 / 0.8466 / 0.9174 / 0.7441 | 0.8744 / 0.8484 / 0.9323 / 0.7918 | 0.8790 / 0.7704 / 0.8440 / 0.7443 | **0.8925** / **0.8558** / **0.9323** / **0.8214** |
>
> **Note:** It is reasonable that QLoRA and LQ-LoRA achieve higher accuracy, as they perform **one-to-one fine-tuning** tailored to specific quantization configurations. In contrast, our proposed CoA-LoRA conducts **one-for-all fine-tuning**. Therefore, achieving performance comparable to QLoRA and LQ-LoRA is expected, while our proposed method also provides the practical benefit of covering all configurations with a single fine-tuning process.
>
> [1] E. Hu, Y. Shen, P. Wallis, et al. LoRA: Low-Rank Adaptation of Large Language Models. ICLR. 2022.
>
> Please refer to Part II for the remaining responses.

---

> ### Author Response · Authors · 2025-11-18
> **Response to Questions (Part II)**
>
> > Question 1: Clarification of "unseen" configuration definition
>
> We clarify that “unseen” refers to configurations that were never included in the entire training configuration set.
>
> > Question 2: Details of the configuration-aware model and explanation of inputs.
>
> **What is a configuration-aware model?**
>
> First, for each layer of the quantized large model, we initialize two low-rank matrices via SVD. Next, the configuration-aware model takes the embedding of each layer’s quantization configuration as input and learns to generate an $r \times r$ low-rank adjustment for each layer. This adjustment allows the model to recover quantization accuracy on downstream tasks.
>
> **Why is it considered light-weight?**
>
> The structure of the configuration-aware model is as follows:
>
> 1. **Input Embeddings:** For each layer, several types of embeddings are constructed:
>
>   - **Module type:** For RoBERTa-Large, the module types include `query`, `key`, `value`, `output.dense`, and `intermediate.dense`; for LLaMA, they include `q_proj`, `k_proj`, `v_proj`, `o_proj`, `gate_proj`, `down_proj`, and `up_proj`.
>   - **Quantization parameters:** $b_0$, $b_1$, $b_2$ (e.g., $\texttt{fp16, bf16, fp32}$), $B_0$, $B_1$.
>   - **Block position**: the block index.
> 2. **Concatenation**: All embeddings are concatenated into a single vector representing the layer and its quantization configuration.
>
> 3. **MLP**: A two-layer MLP with a normalization layer and SiLU activation maps the concatenated vector to an output of size $r×r$ (where $r$ is the LoRA rank). The MLP produces an adjustment matrix for the LoRA parameters of the corresponding layer.
>
> 4. **Output adjustment**: The output matrix is scaled by a learnable parameter and an identity matrix is added to stabilize the transformation, resulting in the final layer-specific adjustment matrix.
>
>
> The model is lightweight because the MLP’s final layer outputs only a small $r \times r$ adjustment matrix for each layer, rather than the full LoRA weight matrices. This ensures that the additional computation and parameter overhead introduced by the configuration-aware model is minimal.
>
> Table R2 presents the parameter count of the configuration-aware mapper in comparison with the total model size and trainable parameters. The configuration-aware mapper accounts for at most 0.5% of the parameters, demonstrating its very low computational overhead.
>
> **Table R2.** Parameter breakdown for different models: total model parameters / LoRA parameters (rank = 128) / configuration-aware mapper parameters.
>
> | Model | Parameters (total / LoRA / mapper) | Percentage (total / LoRA / mapper) |
> | --- | --- | --- |
> | RoBERTa-Large | 414,934,577 / 60,845,617 / 2,121,265 | 100% / 14% / 0.5% |
> | Qwen2.5-1.5B | 1,693,553,729 / 149,839,425 / 2,121,281 | 100% / 8% / 0.1% |
> | Qwen2.5-3B | 3,327,528,513 / 241,589,825 / 2,121,281 | 100% / 7% / 0.06% |
> | Llama-2-7B | 7,060,352,577 / 321,936,961 / 2,121,281 | 100% / 4% / 0.03% |
>
> **How exactly to combine $\boldsymbol{[z, m, b]}$ with $\mathbf{U}$?**
>
> The module type (e.g., query, key, value) embedding corresponds to $\boldsymbol{m}$.
> The embedding of discrete quantization configurations (i.e., $b_0, b_1, b_2, B_0, B_1$) corresponds to $\boldsymbol{z}$. The block position embedding corresponds to $\boldsymbol{b}.\mathbf{U}_{\boldsymbol{\theta}}$ is a model, where $\mathbf{U}$ is a function parameterized by $\boldsymbol{\theta}$.
>
> We concatenate the embeddings $\boldsymbol{z}, \boldsymbol{m}$, and $\boldsymbol{b}$ into a single vector and use this concatenated vector as the input to $\mathbf{U}_{\boldsymbol{\theta}}$.
>
> Due to character limitations, please refer to Part III for the remaining responses.

---

> ### Author Response · Authors · 2025-11-18
> **Response to Questions (Part III)**
>
> > Question 3: Clarification on continuous optimization within a discrete search space.
>
> We thank the reviewer for this comment. We realized that part of the confusion arises from our notation, which we have clarified in the revised manuscript: discrete quantization configurations are now denoted as $\mathbf{C}$, while their corresponding continuous embeddings are denoted as $\mathbf{Q}$.
>
> **Validity of update in Eq. (8):**
> The update in Eq. (8) is based on
> $
> \operatorname{sign}(\frac{\partial \alpha _ {\mathrm{EHVI}}}{\partial \mathbf{C} ^ {(j)} _ {i ^ *}}) \mathbf{e} _ {i ^ *}.
> $
>
>
> Using the $\operatorname{sign}(\cdot)$ function ensures that $\mathbf{C}$ moves along discrete steps of size 1. If an update would move $\mathbf{C}$ outside the valid range specified in Table 1 of the manuscript, we simply skip that update. This guarantees that $\mathbf{C}$ remains a valid discrete configuration at all times.
>
> **Computation of $f_1(\mathbf{C})$ (Eq. 4):**
> To evaluate $f_1(\mathbf{C})$, we first map the discrete configuration $\mathbf{C}$ to its corresponding continuous embedding $\mathbf{Q}$. The configuration-aware model then generates low-rank adjustments for all layers. The low-rank adjustments are first incorporated into the LoRA modules, which are then merged with the quantized base model. The final model is evaluated on a small portion of the training dataset to compute its loss value.
>
> **Computation of $f_2(\mathbf{C})$:**
> The function $f_2(\mathbf{C})$, representing the average bit-width under the layer-wise parameter distribution, is computed as
>
> $$
> f_2(\mathbf{C}) = \frac{\sum_{i=1}^{L} (b_0+\frac{b_1}{B_0}+\frac{b_2}{B_0B_1}) \cdot w_i}{\sum_{i=1}^{L} w_i},
> $$
>
> where $w_i$ is the number of parameters in layer $i$ and $\mathbf{C}=[b_0,b_1,b_2,B_0,B_1]$. Here, $b_2$ denotes the bitwidth of the precision type. If the type is $\texttt{fp16}$ or $\texttt{bf16}$, we set $b_2 = 16$; if it is $\texttt{fp32}$, we set $b_2 = 32$.
>
> With this clarified notation, both $f_1$ and $f_2$ are now unambiguous and well-defined for any discrete configuration $\mathbf{C}$.

---

### Official Review · Reviewer_2MdX · 2025-10-30

**Soundness:** 3
**Presentation:** 2
**Contribution:** 3
**Rating:** 6
**Confidence:** 4

**Summary:**

The authors in this work propose CoA‑LoRA, a configuration‑aware approach that lets a single fine‑tuned adapter serve many quantization settings without re‑finetuning. Their approach conditions a lightweight adapter transformations on the quantization profile and integrates an Pareto configuration search to train over a diverse yet optimal configuration set. Experiments on GLUE and reasoning benchmarks show that CoA-LoRA maintains accuracy comparable to or better than per-configuration baselines while greatly reducing finetuning time and generalizing to unseen quantization settings. The paper’s claims are overall well-supported by results. The methodology is clearly described, and is easy to follow. The experimental setup tested on both GLUE and C4 and the ablations hints towards generalization of their claims. Reporting of training budgets is transparent and helps in comparing fairly. The authors could benefit from describing more detail on the mapper’s architecture and inference latency. This would help assess their proposed implementation's real-world feasibility. The paper is well‑structured and not too difficult to follow, with pseudocode in the appendix. Prior work is discussed appropriately. As an improvement, authors can consider providing more insights into mapper architecture’s detail for better reproducibility. The authors solve a timely and practical problem which if generalizable can help the broader community in efficiently finetuning quantized models. Through experiments, their design choice is motivated by an insightful observation that shared knowledge across configurations is mostly captured in L2​, while configuration-specific information resides in L1​. This is a new insight and  justifies adapting L2 through the lightweight mapping model.

**Strengths:**

1. The authors address an important practical problem of fine tuning of quantized models where per-configuration is not feasible. Their proposed method learns a mapping model which eliminates repeated adapter training while reducing the performance loss due to quantization.

2. Their proposition is based on an important observation of L2 capturing configuration specific information. This is a new insight and justifies adapting L2 through a lightweight transformation. If generalized well, this could be very insightful for the community for future quantized finetuning works.

3. The EHVI‑driven, finite‑difference coordinate search coupled with Pareto filtering is well‑motivated for the accuracy–bit trade‑off.
The authors also demonstrate competitive performance across GLUE and commonsense reasoning tasks.

**Weaknesses:**

1. The paper has limited discussions on the architecture and parameter count of the configuration‑aware mapper ​ and the overhead of generating mapping for a new configuration. Including these discussions would clarify deployability and provide deeper insights to their method for larger audience adaptability.

2. The method is evaluated solely on the NormalFloat (NF) quantization space and does not explore standard integer (INT) quantization schemes such as INT8 or INT4. Demonstrating that CoA-LoRA can generalize to widely used integer quantizers would strengthen the claim of broad applicability.

3. As highlighted in prior work such as QLoRA, the performance gap between pre- and post-fine-tuning typically widens as model size increases. Since CoA-LoRA is evaluated only up to 7B parameters, it remains unclear how the method scales to larger models. Including ablations on 14B-scale or higher architectures would provide valuable insight into whether the proposed configuration-aware adaptation maintains stability and efficiency at greater capacity.
4. The observation that L2​ captures shared knowledge across quantization configurations, while L1​ encodes configuration-specific information, is derived only from experiments on RoBERTa. It remains unclear whether this property generalizes to larger autoregressive or decoder-only architectures such as LLaMA or Qwen. A cross-model analysis of the ​ decomposition would help confirm whether the same structural separation holds beyond encoder-style models.

**Questions:**

1. The L1 and L2 separation is shown only on RoBERTa. Does this pattern hold for decoder-only models like LLaMA or Qwen? Also, does this characteristic change when the rank of the LoRA adapter is increased?

2. How does CoA-LoRA perform with integer quantization (INT4) instead of NormalFloat?

3. Can the method scale to 14B+ models, and what computational limits arise in training or search?

4. Does the configuration-aware mapping support or extend naturally to mixed-precision quantization, or is it limited to homogenous quantization of the model?

---

> ### Author Response · Authors · 2025-11-18
> **Response (Part I)**
>
> We sincerely thank the reviewer for the insightful comments. Our responses to the weaknesses and questions are provided below:
>
> > Weakness 1: Clarification on mapper design and computational overhead
>
> The structure of the configuration-aware model is as follows:
>
> 1. **Input Embeddings:** For each layer, several types of embeddings are constructed:
>
>   - **Module type:** For RoBERTa-Large, the module types include `query`, `key`, `value`, `output.dense`, and `intermediate.dense`; for LLaMA, they include `q_proj`, `k_proj`, `v_proj`, `o_proj`, `gate_proj`, `down_proj`, and `up_proj`.
>   - **Quantization parameters:** $b_0, b_1, b_2$ (e.g., $\texttt{fp16, bf16, fp32}), B_0, B_1$.
>   - **Block position**: the block index.
> 2. **Concatenation**: All embeddings are concatenated into a single vector representing the layer and its quantization configuration.
>
> 3. **MLP**: A two-layer MLP with a normalization layer and SiLU activation maps the concatenated vector to an output of size $r×r$ (where $r$ is the LoRA rank). The MLP produces an adjustment matrix for the LoRA parameters of the corresponding layer.
>
> 4. **Output adjustment**: The output matrix is scaled by a learnable parameter and an identity matrix is added to stabilize the transformation, resulting in the final layer-specific adjustment matrix.
>
>
> The Table R1 presents the parameter count of the configuration-aware mapper in comparison with the total model size and trainable parameters. The configuration-aware mapper accounts for at most 0.5% of the parameters, demonstrating its very low computational overhead.
>
> We have added details of the model architecture and parameter count in Appendix A.1 of the revised manuscript.
>
> **Table R1.** Parameter breakdown for different models: total model parameters / LoRA parameters (rank = 128) / configuration-aware mapper parameters.
>
> | Model | Parameters (total / LoRA / mapper) | Percentage (total / LoRA / mapper) |
> | --- | --- | --- |
> | RoBERTa-Large | 414,934,577 / 60,845,617 / 2,121,265 | 100% / 14% / 0.5% |
> | Qwen2.5-1.5B | 1,693,553,729 / 149,839,425 / 2,121,281 | 100% / 8% / 0.1% |
> | Qwen2.5-3B | 3,327,528,513 / 241,589,825 / 2,121,281 | 100% / 7% / 0.06% |
> | Llama-2-7B | 7,060,352,577 / 321,936,961 / 2,121,281 | 100% / 4% / 0.03% |
>
> > Weakness 2: Evaluation restricted to NF quantization
>
> We conducted additional experiments under integer mixed-precision quantization, where each layer can be assigned $\texttt{int2}$, $\texttt{int3}$, $\texttt{int4}$, or $\texttt{int8}$. The results are shown in the Table R2 . CoA-LoRA consistently demonstrates strong performance across all four tasks, achieving the best results in 10 out of the 16 task–bit combinations.
>
> The experiments and detailed analysis have been added in Section 5.3 of the revised manuscript.
>
> **Table R2:** Accuracy comparison under integer mixed-precision quantization (per-layer choices: $\texttt{int2}, \texttt{int3}, \texttt{int4}, \texttt{int8}$). Columns correspond to methods, and values within each cell are reported in the order of tasks: QNLI / MNLI / SST-2 / QQP.
>
> | Avg. Bit | QLoRA | LQ-LoRA | Shared-LoRA | CoA-LoRA |
> | --- | --- | --- | --- | --- |
> | 3   | 0.8537 / **0.8153** / 0.8291 / 0.7353 | 0.8552 / 0.7397 / **0.9208** / 0.7777 | 0.7772 / 0.6618 / 0.8199 / 0.7311 | **0.8616** / 0.8099 / 0.8589 / 0.7621 |
> | 4   | 0.8828 / **0.8491** / 0.9116 / 0.7469 | 0.8762 / 0.7913 / 0.9185 / 0.7952 | 0.8762 / 0.7632 / 0.8486 / 0.7626 | **0.8841** / 0.8441 / **0.9254** / **0.8223** |
> | 5   | 0.8859 / **0.8554** / 0.9105 / 0.7501 | 0.8761 / 0.8481 / 0.9369 / 0.8000 | 0.8777 / 0.7674 / 0.8383 / 0.7403 | **0.8863**/ 0.8533 / 0.9346 / **0.8160** |
> | 6   | 0.8833 / 0.8466 / 0.9174 / 0.7441 | 0.8744 / 0.8484 / 0.9323 / 0.7918 | 0.8790 / 0.7704 / 0.8440 / 0.7443 | **0.8925** / **0.8558** / **0.9323** / **0.8214** |
>
> Please refer to Part II for the remaining responses.

---

> ### Author Response · Authors · 2025-11-18
> **Response (Part II)**
>
> > Weakness 3: Evaluation on models larger than 7B parameters is not included.
>
> We evaluated CoA-LoRA on LLaMA-3-14B under various quantization bits. Table R3 shows that CoA-LoRA achieves performance comparable to full fine-tuning across most settings, indicating effective adaptation to quantized layers even in larger models (14B).
>
> **Table R3.** Perplexity comparison under different quantization bits (2.5, 3, 3.5, 4) on LLaMA-3 14B.
>
> | Avg. Bit | QLoRA | LQLoRA | GPTQ-LoRA | Shared-LoRA | CoA-LoRA |
> | --- | --- | --- | --- | --- | --- |
> | 2.5 | 9.11 | 9.23 | -   | 8.87 | **8.65** |
> | 3   | **7.48** | 8.00 | 7.95 | 8.18 | 7.58 |
> | 3.5 | **7.09** | 7.52 | -   | 7.59 | 7.15 |
> | 4   | **6.89** | 7.32 | 7.15 | 7.28 | 6.99 |
>
> **Note:** It is reasonable that QLoRA and LQ-LoRA achieve higher accuracy, as they perform **one-to-one fine-tuning** tailored to specific quantization configurations. In contrast, our proposed CoA-LoRA conducts **one-for-all fine-tuning**. Therefore, achieving performance comparable to QLoRA and LQ-LoRA is expected, while our proposed method also provides the practical benefit of covering all configurations with a single fine-tuning process.
>
> > Weakness 4: Cross-architecture consistency of low-rank factorization patterns (shared knowledge and configuration-specific information).
>
> Based on the reviewer’s suggestion, we provide correlations of two types of low-rank matrice similarity in Appendix Fig. C.1 and C.2.
>
> Fig. C.1 shows the results for RoBERTa under different ranks, and Fig. C.2 presents results for Qwen and LLaMA architectures. Across all experiments, a consistent pattern is observed: $\mathbf{L}_1$ captures shared knowledge (high similarity scores), while $\mathbf{L}_2$ focuses on configuration-specific knowledge (low similarity scores).
>
> > Question 1: Evaluation of L2/L1 similarity on different model architectures remains pending.
>
> Please refer to our response to Weakness 4.
>
> > Question 2: How does CoA-LoRA perform with integer quantization (INT4) instead of NormalFloat?
>
> Please refer to our response to Weakness 2.
>
> > Question 3: Can the method scale to 14B+ models, and what computational limits arise in training or search?
>
> Please refer to our response to Weakness 3.
>
> > Question 4: Does the configuration-aware mapping support or extend naturally to mixed-precision quantization
>
> Our configuration-aware mapping naturally supports mixed-precision quantization. In fact, all our experiments are conducted under this setting. Specifically, our method performs layer-wise (heterogeneous) quantization, allowing different layers to be quantized to different bit widths. For all experiments in this work, each layer can be quantized to one of four bit widths: 2, 3, 4, or 8 bits.

---

### Official Review · Reviewer_TGfV · 2025-10-31

**Soundness:** 3
**Presentation:** 2
**Contribution:** 3
**Rating:** 6
**Confidence:** 5

**Summary:**

This paper tackles the issue that LLM models should be quantized multiple times if they are deployed on various devices.
It's because optimal quantization configurations differ according to target hardware.
To resolve this problem, the paper proposes CoA-LoRA that can find optimized LoRA adapters for quantization on-the-fly.

The key idea is to yield adjustment matrices that are used to reparameterize a LoRA adapter.
By generating an adjustment matrix to a target quantization configuration, an optimized LoRA adapter can be achieved.

Adjustment matrices are generated with Configuration aware model that is trained for yielding an adjustment matrix for each layer of LLM.
This model gets a quantization configuration as an input and returns adjustment matrices optimized for the configuration.

In order for this method to work well, Configuration aware model should be trained well with representative configuration settings.
This paper searches configurations with a Pareto-based Gaussian process and finite-difference gradient approximation for training Configuration aware model.
Starting from an initial configuration set, Pareto improved solutions against bit or loss are replaced or added to the configuration set.

With these proposals, the paper designs optimized LoRA adapters for various configuration settings.

**Strengths:**

- This paper aims to run an LLM on various quantization configurations caused by different hardware properties with minimal performance degradation.
Designing LoRA adapters per configuration is too costly, while using a single LoRA adapter on various bit settings leads to performance degradation.
This paper accurately recognizes these problems and proposes a solution.

- It is novel that distributes LoRA adapters on-the-fly by yielding adjustment matrices tailored for quantization configuration.
The proposed configuration search also helps train Configuration aware model to achieve better solutions.

- The paper shows the effectiveness of the proposal by executing various experiments with several datasets, model sizes, and LoRA ranks, in addition to an ablation study about configuration search.

**Weaknesses:**

- It is not clear that the experimental settings of other methods are exactly the same per layer, or only the average bits are set to the same.
Sensitivities of layers to performance can differ, and the performance degradation can be caused by it.
If the experiments are executed with exactly the same configurations, please clarify it.

- Figure 5 is in the main manuscript, but it is not referenced.

- Q-LoRA and LQ-LoRA are costly to execute with the whole bit settings, but the reviewer think that settings over 4-bits should be executed too, to compare the performances on higher bit settings.

- Recently, there have been several works that generate LoRA adapters according to conditions [1, 2]. However, the paper doesn't mention any of them. Discussing them in the final version of the paper would strengthen the paper.

[1] Shao, Yihua, et al. "In-Context Meta LoRA Generation." arXiv preprint arXiv:2501.17635 (2025).

[2] Xiao, Yicheng, et al. "LoRA-Gen: Specializing Large Language Model via Online LoRA Generation" International Conference on Machine Learning. PMLR, 2025.

**Questions:**

- In Figure 5, CoA-LoRA shows better results than full fine-tuning on QQP. The reviewer think that full fine-tuning is more likely to show better results compared to a quantized one.
Can the authors provide additional explanation about that?

- The proposed method facilitates on-the-fly adjustment of LoRA adapters. However, the previous works, such as LQ-LoRA adapts LoRA adapters to a single target setting directly.
Thus, the reviewer think that those direct adaptations are more likely to show better results on a specific target setting.
But CoA-LoRA outperforms other previous works even in their target settings. Can the authors provide your opinions on how this came out?

- As far as the reviewer understand, adjustment matrices are multiplied by a shared LoRA adapter. If so, how could the LoRA adapter, before adjustment can be achieved?

---

> ### Author Response · Authors · 2025-11-18
> **Response to Weaknesses (Part I)**
>
> We sincerely thank the reviewer for the valuable suggestions and comments. We provide the following point-by-point response.
>
> > Weakness 1: Layer-wise bit configuration
>
> For a fair comparison, other methods such as QLoRA and LQ-LoRA adopt layer-wise bit quantization rather than using the same bit-width for all layers. Given an average bit-width, these methods treat the per-layer quantization bits as optimization variables and aim to minimize the discrepancy between the quantized model and the original model, subject to the constraint of the specified average bits. The detailed mathematical formulation can be found in Eq. (11) in the appendix of the manuscript.
> Based on the reviewer’s suggestion, we have added a clarification of this setting in Section 5.1 of the revised manuscript.
>
> > Weakness 2: Unreferenced Figure 5
>
> An explanation of Figure 5 has been included in Section 5.2 of the revised manuscript.
>
> > Weakness 3: High-bit comparison requested for Q/LQ-LoRA.
>
> We have added higher-bit results (i.e., 5- and 6-bit) for QLoRA and LQ-LoRA in Fig. 5 of the revised manuscript. The results show that our proposed CoA-LoRA achieves comparable or even better performance under these settings.
>
> For clarity, the updated experiments under 5- and 6-bit settings are summarized below. Each cell shows the results for 5-bit and 6-bit configurations separated by a slash; for example, “0.8665/0.8931” indicates that the result is 0.8665 for 5-bit and 0.8931 for 6-bit.
>
> | Method | QNLI | MNLI | SST-2 | QQP |
> | --- | --- | --- | --- | --- |
> | QLoRA | 0.8665/0.8931 | 0.8538/0.8554 | 0.9392/0.9311 | 0.8432/0.8425 |
> | LQ-LoRA | 0.8374/0.8815 | 0.8451/0.8416 | 0.9311/0.9403 | 0.7892/0.8225 |
> | Shared-LoRA | 0.8405/0.8407 | 0.8133/0.8132 | 0.8521/0.8463 | 0.7147/0.7005 |
> | CoA-LoRA | 0.8694/0.8709 | 0.8315/0.8318 | 0.9266/0.9334 | 0.8416/0.8372 |
>
> **Note:** It is reasonable that QLoRA and LQ-LoRA achieve higher accuracy, as they perform **one-to-one fine-tuning** tailored to specific quantization configurations. In contrast, our proposed CoA-LoRA conducts **one-for-all fine-tuning**. Therefore, achieving performance comparable to QLoRA and LQ-LoRA is expected, while our proposed method also provides the practical benefit of covering all configurations with a single fine-tuning process.
>
> > Weakness 4: Missing discussion of conditionally generated LoRA adapters.
>
> We would like to clarify that the way we adjust LoRA in our work is fundamentally different from the two works mentioned by the reviewer ([1,2]):
>
> - **Computational efficiency:** Both of the cited works require generating a training dataset containing well-trained or expert LoRAs to learn task-specific embeddings, which is computationally expensive—each instance typically takes from **half an hour to several hours**. In contrast, our method does not rely on such costly resources. Given an average bit-width, we efficiently compute the layer-wise configuration set using the optimization function in Eq. (11) in the appendix via Gurobi, with each instance taking only **3–5 seconds**.
>
> - **Storage and latency efficiency:** The LoRA generation paradigms in prior works are computationally and memory intensive: LoRA-Gen [2] requires loading a large number of expert LoRAs for combination, while ICM-LoRA [1] generates each parameter through multi-step diffusion, resulting in high latency. In contrast, CoA-LoRA embeds a conditionally learnable $r \times r$ matrix before $L_2$, enabling **lightweight parallel generation of all layers’ matrices with minimal memory overhead**. This approach allows efficient inference and storage, making it significantly more resource-friendly than existing methods.
>
> - **Application domain:** Our work is the first to adjust LoRA in an efficient manner under different quantization settings. The methods in [1,2] cannot be directly applied to this scenario because they rely on expert LoRA knowledge and many well-trained LoRAs, which are difficult to obtain for large models.
>
> We have added references and discussion of these LoRA generation works in Section 2.3 of the revised manuscript.

---

> ### Author Response · Authors · 2025-11-18
> **Response to Questions (Part II)**
>
> > Question 1: Unexpected QQP results: CoA-LoRA vs. full fine-tuning
>
> Although full fine-tuning is theoretically superior to LoRA, in practice LoRA can occasionally match or even surpass full fine-tuning, depending on the task. For example, in [3], LoRA outperforms full fine-tuning on COCO Captioning (Table 2 in the original paper), and similarly on visual instruction tuning tasks (see Table 3). Fig. 1 of [4] also shows that, with appropriate rank settings, LoRA can sometimes exceed full fine-tuning. Moreover, in the LQ-LoRA paper, it occasionally outperforms full fine-tuning (Fig. 2 in the original paper).
>
> This phenomenon can be partly explained by task characteristics and model generalization. QQP dataset is a text similarity task where some examples have subtle semantic differences, making them more challenging for the model to generalize. Full fine-tuning updates all model parameters, which can fit training data very closely, while LoRA updates only low-rank adapters. This constrained update acts as a form of implicit regularization, reducing sensitivity to subtle variations and potentially improving generalization on challenging examples. As a result, CoA-LoRA can occasionally outperform full fine-tuning on QQP. The reason that CoA-LoRA is more likely to outperform full fine-tuning on QQP, compared to other methods, is explained in our response to Question 2.
>
> [3] S. Liu, C. Wang, H. Yin, et al. Dora: Weight-Decomposed Low-Rank Adaptation. International Conference on Machine Learning. 2024.
>
> [4] D. Biderman, J. Portes, J. J. G. Ortiz, et al. LoRA Learns Less and Forgets Less. Transactions on Machine Learning Research. 2024.
>
> > Question 2: Understanding CoA-LoRA’s superior performance over LQ-LoRA
>
> There are two main reasons why CoA-LoRA can outperform QLoRA and LQ-LoRA.
>
> - As shown by the similarity matrices in Fig. C1 and Fig. C2 of the appendix, $\mathbf{L}_1$ captures shared knowledge across different average quantization configurations (i.e., high similarity), while $\mathbf{L}_2$ focuses more on configuration-specific knowledge for each bit setting. This observation indicates that LoRA fine-tuning under different average bits exhibits similar patterns rather than being entirely unrelated tasks. CoA-LoRA, by training on a diverse set of quantization configurations, effectively leverages this shared structure, thereby enhancing the learning of shared knowledge.
>
> - High-bit configurations preserve more information from the original full-precision model, providing more reliable gradient directions for learning. Low-bit configurations can then take advantage of the parameter patterns (or structures) learned under high-bit conditions to efficiently converge to suitable LoRA parameters, rather than learning from scratch as in QLoRA and LQ-LoRA. This effect is illustrated in Fig. 5 of the revised manuscript: under low-bit settings, CoA-LoRA often outperforms QLoRA and LQ-LoRA, while in high-bit settings, QLoRA and LQ-LoRA can achieve performance comparable to CoA-LoRA. This indicates that low-bit training provides little benefit for high-bit configurations.
>
>
> > Question 3: Clarification on obtaining the shared LoRA adapter prior to adjustment.
>
> The LoRA adapter is **not shared across layers or different quantized models**; each adapter is initialized independently for its specific layer and quantization configuration, so “shared” only occurs if the configuration is exactly the same.
>
> Detailed procedures for the initialization can be found in Appendix A.2.

---

> > ### Comment · Reviewer_TGfV · 2025-11-27
> >
> > The reviewer thanks the authors for their sincere responses to the reviewer’s questions.
> >
> > However, it seems that there is a misunderstanding regarding Question 3.
> >
> > As far as the reviewer understands, the LoRA adapters are first initialized, and then the adjustment matrices are multiplied with the LoRA adapters according to the quantization configuration.
> >
> > That is, it is the reviewer’s understanding that the LoRA adapters are initialized for a single model, and that the adjustment matrices are applied depending on the quantization configuration.
> >
> > From this point of view, the reviewer wonders how one can obtain the base LoRA adapters before the adjustment matrices are applied.
> >
> > The question did not imply that LoRA adapters are shared across layers or across different models.
> >
> > Therefore, the reviewer would like to clarify the question as follows:
> > - Given a target model A, there exists a single base LoRA adapter for each layer.
> > - Depending on the quantization configuration, specific adjustment matrices are multiplied with these LoRA adapters.
> > - If this is correct, how can we obtain the base LoRA adapter? (e.g., is there a default quantization-bit setting used to derive the base LoRA adapter?)
> >
> > Please correct the reviewer if there is any misunderstanding.

---

> > > ### Author Response · Authors · 2025-11-28
> > >
> > > Thank you very much for the reviewer’s follow-up.
> > >
> > > It seems that the reviewer’s comment assumes that all quantized models share a single base LoRA adapter set. However, in our method, **there is no default or shared base LoRA adapter**. Instead, the LoRA adapters are specific to each quantization configuration and are obtained by applying SVD to the difference between the quantized weights and the original pretrained weights.
> > >
> > > In particular, given the quantization configuration of a layer, our algorithm first performs SVD on the residual between the quantized weight and the pretrained weight to obtain the LoRA adapter. We then generate the corresponding LoRA adjustment **for that quantization configuration**. In other words, deriving the LoRA adapter via SVD is an internal step of our algorithm — it is **not** an initialization of a default base adapter.
> > >
> > > To make this clearer, we have updated Equation (2) and the accompanying explanation in the revised manuscript.
> > >
> > > We hope this clarifies the misunderstanding. We sincerely appreciate the reviewer’s comments and feedback.

---

### Author Response · Authors · 2025-12-01
**Summary of Our Responses to Reviewers' Concerns**

Dear ACs, SACs, and PCs,

Thank you for handling our submission. We appreciate the reviewers’ recognition of our work’s "novelty", "effectiveness", and "practical relevance", as well as their comments noting that it is "insightful for the community" and "well-motivated". We also thank reviewers for the constructive suggestions and feedback. Below is a concise summary of the contributions of our work and how our rebuttal addresses the key concerns.

---

**Contributions**

We propose CoA-LoRA, a configuration-aware approach that generates lightweight and quantization-aware low-rank adjustments to LoRA adapters without repeated fine-tuning. Experiments demonstrate that CoA-LoRA performs consistently across bit-widths from 2.5 to 6.5, model sizes from 0.3B to 14B, and multiple datasets, making it practical for efficient and flexible deployment of large language models.

---

**Method-related concerns.**

Reviewers raised questions regarding (1) whether our framework can account for hardware-related constraints such as latency, (2) the computational overhead of our configuration-aware mapper, (3) implementation details, and (4) reproducibility.

We clarified that our approach **naturally accommodates hardware-specific configurations**, as users can freely select quantization settings supported by their hardware, and our method can automatically provide the corresponding LoRA adjustments to recover quantization accuracy. We further showed that our proposed method introduces **less than 0.5% additional parameters**, and we expanded the description of our method by providing detailed information about the module’s structure. To address reproducibility, we offered detailed implementation clarifications and released anonymized code.


**Experiment-related concerns.**

Reviewers requested (1) explanations of several experimental phenomena, (2) higher-bitwidth results, (3) extensions from NormalFloat to mixed integer quantization, (4) evaluations on larger models, and (5) clarification regarding potential dataset-size effects in the configuration-search ablation study.

In our rebuttal, we provided detailed explanations of the observed behaviors and added experiments covering higher-bit settings, mixed-precision integer quantization, and scaling to a 14B model. These results consistently support our method’s effectiveness across different settings. For the configuration-search ablation study, we conducted additional analysis experiments to verify that the observed gains stem from the proposed selection strategy rather than merely from enlarged the training configuration set.

During the discussion period, two reviewers provided responses to our rebuttal. Reviewer Daov confirmed that the raised concerns were fully addressed and requested an additional experiment, which we have included in our updated rebuttal; **the review score for this reviewer was subsequently updated from 4 to 8**. Reviewer TGfV noted that most concerns were addressed and raised a small follow-up question, which we have clarified.

We have updated the manuscript with the new experimental results and the revisions suggested by the reviewers.

Thank you for your time and consideration.

Best regards,

The Authors

---

### Meta-Review · Area_Chair_JEvR · 2026-01-19

**Summary:**

This paper tackles the problem of repeated quantization of the same LLM model due the need to deploy it on devices of various capabilities which require different quantization configurations. To resolve this problem, the paper proposes CoA-LoRA - a methodology that can find optimized LoRA adapters for quantization on the fly. The key idea is to use adjustment matrices that are used to re-parameterize the LoRA adapter. By generating an adjustment matrix to a target quantization configuration, an optimized LoRA adapter can be searched for and achieved. Adjustment matrices are generated with a configuration-aware model that is trained to yield an adjustment matrix for each layer of an LLM. This model takes the quantization configuration as an input, and returns the adjustment matrices optimized for the configuration. In order for this method to work well, a configuration aware model should be trained well with representative configuration settings. This paper searches configurations with a Pareto-based Gaussian process and finite-difference gradient approximation. Starting from an initial configuration set, Pareto improved solutions against bit or loss are replaced or added to the configuration set.

**Reviewer Concerns:**

All but minor concerns were sufficiently addressed in the rebuttal. Still, I recommend the authors to incorporate these responses in the camera-ready version of the paper.

**Reviewer Scores:**

Reviewer Daov planned to increase his/her score from 4 to a higher score. This review was involved in a substantial discussion with the authors. The authors claim the reviewer was planning to increase the score to 8. I do not find evidence for this claim, but the exact value does not matter as, with all other scores included, the paper in the acceptance category in my mind anyway.

I believe that the remaining 3 reviewers would have kept their scores at 6.

---

### Decision · Program_Chairs · 2026-01-26

Accept (Poster)